# Summer aerosol measurements over the East Antarctic seasonal ice zone.

Jack B. Simmons[1], Ruhi S. Humphries[2,3], Stephen R. Wilson[1], Scott D. Chambers[4], Alastair G. Williams[4], Alan D. Griffiths[4,1], Ian M. McRobert[5], Jason P. Ward[2], Melita D. Keywood,[2,3] Sean Gribben[2]

[1]Centre for Atmospheric Chemistry, School of Earth, Atmospheric and Life Sciences, University of Wollongong, Wollongong NSW 2522, Australia
[2]Climate Science Centre, CSIRO Oceans and Atmosphere, Aspendale VIC 3195, Australia
[3] Australian Antarctic Program Partnership, University of Tasmania, Hobart TAS, Australia
[4] ANSTO, Environmental Research, Locked Bag 2001, Kirrawee DC NSW 2232, Australia
[5] Engineering and Technology Program, CSIRO National Research Collections Australia, Hobart TAS 7004, Australia

*Correspondence to*: Jack Simmons (js828@uowmail.edu.au)

**Abstract.** Aerosol measurements over the Southern Ocean have been identified as critical to an improved understanding of aerosol-radiation and aerosol-cloud interactions, as there currently exists significant discrepancies between model results and measurements in this region. The atmosphere above the Southern Ocean provides crucial insight into an aerosol regime relatively free from anthropogenic influence, yet it's remoteness ensures atmospheric measurements are relatively rare. Here we present observations from the PCAN campaign, hosted aboard the *RV Investigator* during a summer (January-March) 2017 voyage from Hobart, Australia to the East Antarctic seasonal sea ice zone. A median particle number concentration (condensation nuclei > 3 nm; $CN_3$) of 354 (95%CI 345-363) cm$^{-3}$ was observed from the voyage. Median cloud condensation nuclei (CCN) concentrations were 167 (95%CI 158-176) cm$^{-3}$. Measured particle size distributions suggested that aerosol populations had undergone significant cloud processing. To understand the variability in aerosol observations, measurements were classified by meteorological variables. Wind direction and absolute humidity were used to identify different air-masses, and aerosol measurements were compared based on these identifications. $CN_3$ concentrations measured during SE wind directions (median 594 cm$^{-3}$) were higher than those measured during wind directions from the NW (median 265 cm$^{-3}$). Increased frequency of measurements from these wind directions suggests the influence of large-scale atmospheric transport mechanisms on the local aerosol population in the boundary layer of the East Antarctic seasonal ice zone. Modelled back trajectories imply different air mass histories for each measurement groups, supporting this suggestion. $CN_3$ and CCN concentrations were higher during periods where the absolute humidity was less than 4.3 $g_{H2O}$/m$^3$, indicative of free tropospheric or Antarctic continental airmasses, compared to other periods of the voyage. Increased aerosol concentration in air-masses originating close to the Antarctic coastline have been observed in numerous other studies. However, the smaller changes observed in the present analyses suggest seasonal differences in atmospheric circulation including lesser impact of synoptic

low-pressure systems in summer. Further measurements in the region are required before a more comprehensive picture of atmospheric circulation in this region can be captured and its influence on local aerosol populations understood.

## 1 Introduction

Aerosol-radiation and aerosol-cloud interactions are among the most uncertain parameters in the current estimates of global radiative forcing (Carslaw et al., 2013, Myhre et al., 2013), and hence a major uncertainty in calculating the climate effects of changing atmospheric composition. In particular, there is a need for reliable estimates for aerosol loadings in the pre-industrial atmosphere, as this is the reference-point for many calculations and the largest contributor to uncertainties in changes in aerosol radiative forcing over the industrial period (Carslaw et al., 2013, Regayre et al., 2014, McCoy et al., 2020). Unlike greenhouse gases, the radiative impact of aerosols cannot be determined from atmospheric archives. However, measurements in pristine remote regions of the Earth can provide an insight into pre-industrial aerosol populations and processes (Carslaw et al., 2017), and have therefore been identified as critical to reducing this uncertainty (Carslaw et al., 2013). Measurements from the Southern Ocean particularly, distant from anthropogenic and continental influence, have been identified as critical to constraining uncertainty in aerosol radiative forcing (Penner et al., 2012, Regayre et al., 2020), and the value of adding Southern Ocean measurements in substantially reducing the uncertainty in estimates of the radiative forcing due to aerosol-cloud interactions (Regayre et al., 2020).

Aerosol optical depth (AOD) is also not well understood in this region. A comparison between models and satellite measurements of AOD show significant underprediction by the models in this region (Shindell et al., 2013) during the summer, possibly related to the modelling of biological aerosol precursors and their chemistry (Revell et al., 2019). It must also be noted satellite measurements are also poorly validated in this region due to the lack of in situ measurements in a region of high cloudiness. Simulated sea surface temperatures (SST) over the Southern Ocean from models participating in CMIP5 (the Coupled Model Intercomparison Project, 5th phase) contain biases attributed to cloud-related error (Hyder et al., 2018), which may relate indirectly to local aerosol populations. Error in modelled SST can propagate through and contribute uncertainty to other parts of global climate models. There is a persistent positive bias in the predictions of absorbed shortwave radiation over the Southern Ocean, the partial correction of which results in increased air mass movement poleward in the Southern Hemisphere (Kay et al., 2015).

There are therefore two significant motivators encouraging *in situ* aerosol measurement campaigns to the remote Southern Ocean: the pristine nature of the local aerosol regime provides an important proxy for an unpolluted pre-industrial atmosphere valuable for assessing aerosol influence on global radiative forcing; and the need to provide further observational constraints in a region of the planet which is difficult to simulate and therefore contributes uncertainty to global climate models.

Surface measurements in the remote Southern Ocean are scarce due to the extreme atmospheric and oceanic conditions, along with the remoteness which defines the region. Longer-term in situ measurements have been made at terrestrial stations on the fringes of the Southern Ocean such as Cape Grim, Tasmania (40.7°S 144.7°E) (Gras and Keywood, 2017), Macquarie Island (54.5°S 158.9°E)

(Brechtel et al., 1998), and Dumont D'Urville Station, Antarctica (66.7°S 140.0°E) (Legrand et al., 2016). There have also been numerous short-term ship-based aerosol measurement campaigns in the region. Ship based measurements taken between 41°S and 54°S observed a bimodal aerosol population
dominated by sea salt (Quinn et al., 1998) during the ACE1 campaign. Sea salt aerosol dominated the mid- and high-latitude Southern Ocean boundary layer aerosol mass concentration, with the aerosol generated by processes such as sea spray and bubble bursting. Voyages to and from Antarctica have also proved valuable. O'Dowd et al. (1997) observed polar and maritime air masses with distinctly different aerosol properties, with sulphate aerosols dominating the polar air mass accumulation mode.
Dominant sulphate contribution to local aerosol populations has also been observed at coastal stations on the Antarctic continent (Shaw, 1988). The source of sulphate in the near-shore region of the remote Southern Ocean is believed to be biological: phytoplankton blooms emit di-methyl sulfide (DMS) which is oxidised in the atmosphere and can eventually condense and grow aerosol (Rinaldi et al., 2010). The distribution of phytoplankton populations is non-uniform and has a strong seasonal signal. A
relationship between methane sulphonic acid (MSA, a DMS oxidation product) and CCN concentration has been observed at Cape Grim (Ayers and Gras, 1991), long with a summer maximum in CCN concentration related to the strength of the phytoplankton DMS source (Gras and Keywood, 2017).The greatest population of these phytoplankton is observed during the spring and summer sea ice melt in areas in the seasonal ice zone (Deppeler and Davidson, 2017). The seasonal sea ice zone, the area of
ocean between the permanent ice edge and winter sea ice maximum, covers a larger spatial area than the Antarctic continent itself.
Aerosol measurements in the seasonal ice zone are therefore of particular interest due to the proximity to the largest secondary aerosol precursor source in the region. Previous measurements in this zone are sparse, though have increased in number in the past decade. Davison et al. (1996) found maximum
levels of DMS and methanesulfonic acid in aerosols in the seasonal sea ice region compared to open oceans further north. Other voyages have observed new particle formation events (Atkinson et al., 2012) and a previously unaccounted source of organic nitrogen in seasonal ice zone aerosol populations (Dall'Osto et al., 2017). These results suggest aerosol populations of the seasonal ice zones are distinct from those of the Antarctic continent and the mid- and upper-latitudes of Southern Ocean. Campaigns
in the last decade, such as SOAP (Law et al., 2017), SIPEXII (Humphries et al., 2015), ACE-SPACE (Schmale et al., 2019), CAPRICORN (Humphries et al., 2021), MARCUS and SOCRATES (Sanchez et al., 2020), have measured aerosol properties in the Antarctic seasonal ice zone from ship-based and aircraft platforms.
Measurements from the East Antarctic seasonal ice zone are particularly scarce, though there is a
growing body of research including measurements from this region. The first results from this region, from the 2012 SIPEXII campaign on board the icebreaker *Aurora Australis* produced surprising results: a step change increase in aerosol number concentration crossing the atmospheric polar front into the Polar cell (Humphries et al., 2016). The atmospheric polar front is the mobile boundary between the Ferrel and Polar cells, the two major circulation cells impacting the Antarctic sea ice zone, located at
latitudes around 60°S. A newly identified circulation mechanism that brought recently formed free-tropospheric secondary aerosols to the Polar cell surface was proposed to explain this observation (Humphries et al., 2016). Alroe et al. (2019) also observed a change in aerosol populations across the atmospheric polar front, attributing the change to increased aerosol precursor concentration in the Polar

cell and free troposphere/boundary layer airmass exchange in synoptic level systems. Aerosol formation
above the marine boundary layer was directly observed above the East Antarctic seasonal ice zone by
Sanchez et al. (2020). They also noted that air masses with the highest particle concentrations observed
during the voyage had recently crossed the Antarctic coast. A recent study ingesting measurements from
multiple voyages to this region of the Southern Ocean reports significantly increased MSA
concentrations in the parts of the Southern Ocean closest to the Antarctic coast, alongside enhancements
in CCN concentration in the latitude band 65-70°S. (Humphries et al., 2021).
A further influence on aerosol properties is the katabatic drainage off the Antarctic continent. Chambers
et al. (2018), using measurements from the presently discussed voyage, observed the influence of
katabatic outflow on ship-based aerosol measurements up to 200 km from the coast. This outflow would
transport air that has been in contact with the continent to regions where free tropospheric air might be
expected.
Here aerosol number concentrations and aerosol size distributions taken during a voyage to the seasonal
ice zone of east Antarctica in the austral summer of early 2017 are presented. Measurements were made
on the *RV Investigator* during the Polar Cell Aerosol Nucleation (PCAN) project. The scientific
motivations of this work are as follows: 1) report on aerosol measurements of the poorly understood and
infrequently measured East Antarctic seasonal ice zone,  2) attempt to better understand the atmospheric
boundary at the polar front observed in this region in multiple recent studies and 3) compare the austral
summer measurements presented here with those taken in spring 2012 in the same location.

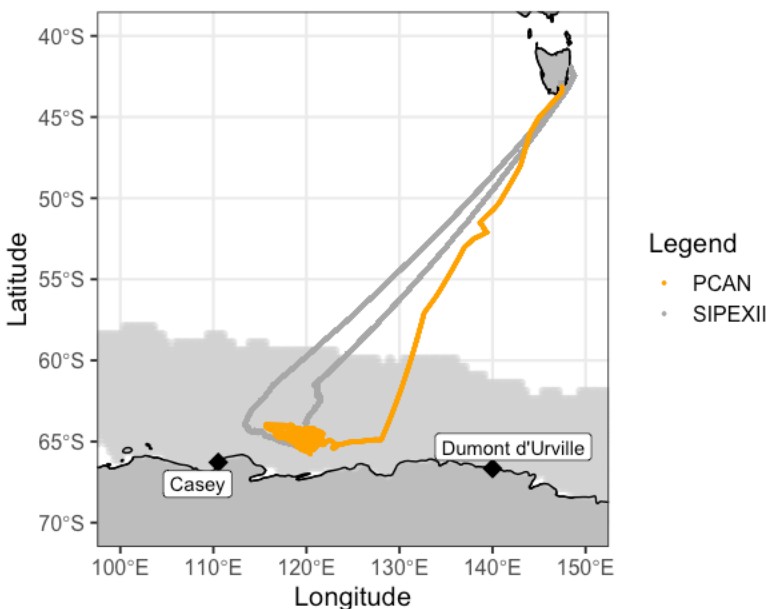

**Figure 1: Voyage tracks of the SIPEX-II and PCAN voyages. Voyage tracks of the SIPEX-II (spring 2012, grey) and periods of the
PCAN voyage (summer 2017, orange) for which measurements were analysed. Measurements from the southward PCAN transit**
**are excluded due to instrument malfunction. Grey shading represents the sea ice area observed at the mid-point of the SIPEXII
voyage as detected by the Nimbus-7 satellite (National Snow and Ice Data Centre, 2021). No sea ice was present in the plotted
region at the midpoint of the PCAN voyage. Nearby Antarctic stations are plotted to assist interpretation. An interactive map of
RV Investigator voyage tracks can be found here: https://www.cmar.csiro.au/data/underway/?survey=in2017_v01.**

## 2 Methods

### 2.1 Measurement platform

Measurements were made aboard the *RV Investigator*, Australia's flagship blue-water research vessel managed by CSIRO's Marine National Facility, during the voyage IN2017-V01. The onboard atmospheric measurements (the Polar Cell Aerosol Nucleation project; PCAN) were a secondary objective of the voyage, the primary focus of which was mapping the bathymetry off the east Antarctic coast and retrieving sediment samples. The voyage departed Hobart, Australia on January 14, 2017. It tracked south-west for six days to the study area, south of 60° S and between 110° and 120° E, where the ship remained for 40 days. The ship returned to port on March 3, 2017 after a seven-day transit northeast. Within the survey area, the ship followed a 'mowing-the-lawn' pattern while seafloor mapping, resulting in high-resolution spatial coverage of a relatively small area. The ship was stationary, facing into the wind, for up to four hours at a time to allow for sediment retrieval. The voyage track, compared to that of the SIPEX-II voyage of 2012, is presented in Figure 1.

### 2.2 Measurements

Wind speed, direction and other meteorological parameters were measured from two locations (port and starboard) on a mast approximately 20 m above the Plimsoll line on the foredeck. This is approximately 25 metres above sea level. Meteorological measurements were made at 1 Hz resolution. Five-minute vector mean wind direction and mean wind speed values were calculated from the two meteorological stations and used for analysis. No bias correction has been applied to correct for any impact of the ship structure. The relative position of the anemometers is forward of the ship's superstructure and exhaust stack (Humphries et al., 2019). Therefore, it is expected that when meteorological measurements may have been impacted by the ship's superstructure the sampled air will carry a ship exhaust signature and have hence been removed by the exhaust filter described below.

The primary sampling inlet, which services both aerosol and greenhouse gas instrumentation, consists of a conic nozzle with an entrance of 41.5 mm diameter, that opens up to a 161.5 mm diameter stainless steel cylinder. The horizontal nozzle is actively directed into the wind with a directional range of 300°. From the nozzle, the inlet bends 90° and continues down vertically for approximately 7.5 m (with two 45° bends), penetrating the deck before reaching a custom designed sampling manifold where the sample line splits into numerous smaller lines, a substantial bypass flow, and a water trap. The resulting sampling height from this setup is approximately 18.4 m above sea level, depending on the ship's draft at any time. The primary flow rate of the sampling line is set at $420 \pm 15$ L min$^{-1}$.

Inlet losses have not yet been experimentally characterised for the aerosol sampling system aboard the *Investigator* but are a point of future work for the platform. Theoretical loss calculations performed for this system suggest >95 % transmission of 10 nm particles through this system and ~75% transmission of 1 nm particles. Reported particle concentrations are uncorrected and therefore represent a lower bound on ambient aerosol population concentration. The present work includes comparison to aerosol concentrations reported by Humphries et al. (2016) taken on board a different research vessel, the *Aurora Australis*. It is prudent at this stage to note the differing inlet efficiencies for this separate inlet

system. An overall inlet transmission efficiency of 0.89 has been reported, generated from a combination of theoretical and experimental characterisation (Humphries et al., 2015).

Number concentrations of in situ aerosols with diameter greater than 3 nm ($CN_3$) were measured using an ultrafine condensation particle counter (UCPC, TSI Model 3776, TSI, Shoreview, MN, USA). Measurements were made at 10 Hz temporal resolution, with 1 Hz averages output by the instrument. Air was sampled at 1.5 L min$^{-1}$ through the UCPC. Zeros (HEPA filtered outside air) and flow checks (external volumetric flowmeter) were made every second day during the voyage, with flow calibrations applied to the measurements in post-processing. A software malfunction led to incorrect times being assigned to measurements during the first part of the voyage. It has not proven possible to reconstruct a reliable time stamp on the early measurements, which means that they cannot be compared to other measurements or reliably filtered for the effects of the ship exhaust. This limited usable $CN_3$ measurements to the period from February 6th to the end of the voyage on March 4th.

Aerosol size distributions (mobility diameter) were measured using a scanning mobility particle sizer (GRIMM nano Model 5.420 SMPS +C, Ainring, Bavaria, Germany). During this campaign particles between 8 and 500 nm were sized. Size distributions were measured at five-minute intervals. Comparison to polystyrene latex spheres showed error in size classification of $\leq 5\%$ for particles of 81, 100 and 303 nm diameter. Sample air for both aerosol number concentration and aerosol size distribution measurements were dried using a Nafion drier (Ecotech, Melbourne, VIC, Australia) in a bypass flow.

Cloud condensation nuclei number concentration was measured using a Cloud Condensation Nuclei Counter (CCN100, Droplet Measurement Technologies, Longmont, Colorado, USA). CCN at 0.55% supersaturation ($CCN_{0.55}$) is reported in this study. Pressure corrections were applied to the data, collected at 1Hz and reported as five-minute means. The sample flow was 0.5 L min$^{-1}$.

$^{222}$Rn (radon) concentration was measured during the voyage using a 700 L dual-flow-loop two-filter radon detector, designed and built by the Australian Nuclear Science and Technology Organisation (Lucas Heights, NSW, Australia) (Chambers et al., 2014; Whittlestone and Zahorowski, 1998) with a flow rate of 65-75 L min$^{-1}$. Calibration occurs on a quasi-monthly basis through radon injections from a Pylon (Ottawa, ON, Canada) $^{222}$Rn source ($20.62 \pm 4\%$ kBq $^{226}$Ra). $Rn^{222}$ was measured as it can be a useful tracer of terrestrial influence on air masses, particularly in the remote marine environment, and thus provide information on the history of sampled air masses.

Black carbon aerosol and $CO_2$ were also measured as part of the permanent instrumentation aboard the RV Investigator. These measurements were only used for exhaust identification, and therefore these data are not presented or discussed here. Black carbon aerosol measurements were made using a Multi-Angle Absorbtion Photometer (MAAP model 5012, Thermo Fisher Scientific, Air Quality Instruments, Franklin, MA, USA). Carbon dioxide is measured using a cavity ring-down spectrometer (Picarro model G2301, Picarro Inc., Santa Clara, CA, USA). Further detailed descriptions of these measurements are available elsewhere (Humphries et al., 2019).

Significant periods of the voyage were affected by contamination of the sampled air by the ship's exhaust due to the operational pattern of the ship. An automated exhaust removal procedure was used on the data using an algorithm designed by Humphries et al. (2019), which identifies periods where measurements are likely to be contaminated by the ship platform's own exhaust, and can be subsequently removed. Measurements of $CN_3$ and $CO_2$ that were greater than the five-minute median

added to three times the median absolute deviation; or containing black carbon concentrations greater than 0.07 ng m$^{-3}$ were flagged as exhaust. Automatically filtered measurements were then manually inspected in conjunction with black carbon and $CO_2$ measurements for signs of contamination, and suspect periods removed. This was done in order to eliminate the influence of re-circulated, diluted exhaust.

As the exhaust filter relies on $CN_3$, it is not possible to filter the data for the first part of the voyage in a manner consistent with the rest of the trip. Additionally, to ensure no bias in a comparison of the $CN_3$ number concentrations with the other quantities measured, it was necessary to limit this analysis to the time period of the shortest dataset, $CN_3$ measurements as noted above (February 6th until the end of the voyage).

## 2.3 Back trajectory modelling

Back trajectories presented were calculated using NOAA's HYbrid Single Particle Lagrangian Integrated Trajectory (HYSPLIT) model (Draxler and Hess, 1998). The ERA-Interim reanalysis dataset (European Centre for Medium-Range Weather Forecasts, 2019) was used to produce trajectories. The lack of meteorological observations at high latitudes means that there is significant uncertainty in the ERA datasets. Path uncertainty of 15-30% of the trajectory travel distance has been previously estimated over the Antarctic continent (Scarchilli et al., 2011). This uncertainty was attributed both to the use of low-resolution meteorological data, and numerical uncertainty inherent in the model. Trajectories were calculated for each hour of the voyage including vertical transport and limited to 72 hours to minimize the accumulation of this uncertainty. The end point was set at the ship's location at 20 m above sea level (approximately sampling height) for each calculation.

## 2.4 Data analysis

Data analysis was performed in the programming language 'R' (R Core Team, 2017). Code used for analysis and to produce figures included in this publication can be supplied upon request. All measurements used for analysis were resampled to five-minute time resolution. For consistency between figures and statistics reported in text, confidence intervals for all reported values except in modal bin sizes were calculated following McGill et al. (1978), scaling the quotient of the interquartile range and square root of the number of observations. The notch width in all box plots is also calculated using this method. Median concentrations are reported with 95% confidence intervals (denoted with CI) unless noted. Reported $r$ values are Pearson correlation coefficients.

## 3 Results and Discussion

In the proceeding sections, categorisation of air masses has been performed using both wind direction and absolute humidity. Following a definition of the borders of the categories, the overall aerosol properties observed during the voyage are presented. Differences in aerosol properties within categories

defined by wind direction and absolute humidity are then considered and the implications of the
findings discussed.

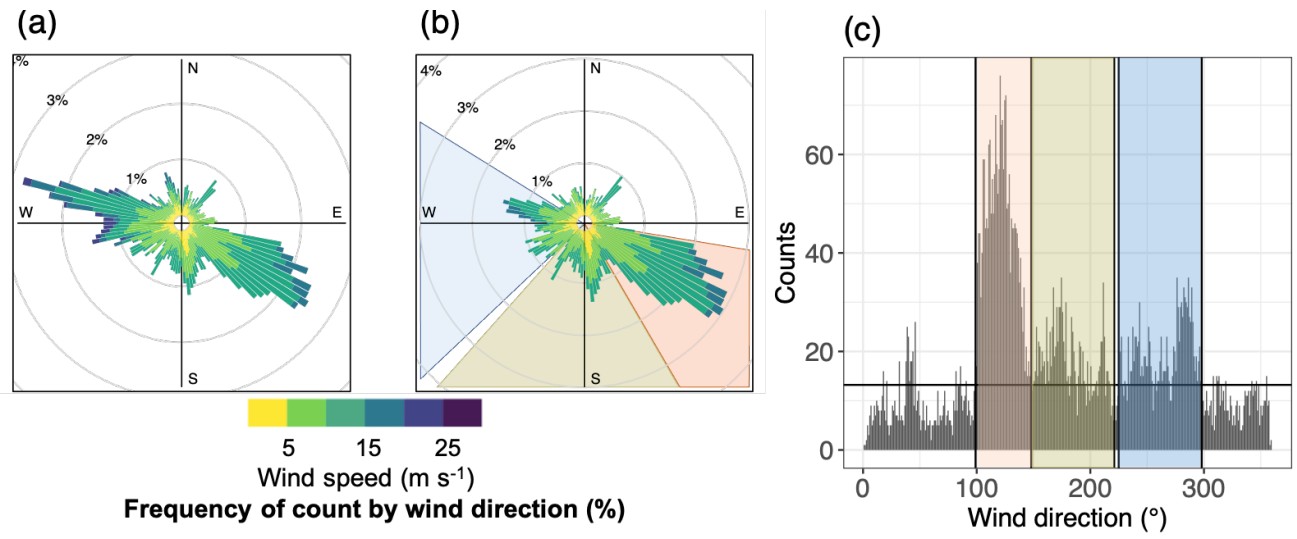

**Figure 2 (a): Wind rose for the entire analysed PCAN voyage (February 6-March 4, 2017). (b): Wind rose for wind measurements south of 56.5° S (excluding measurements from the homeward transit to Hobart Tasmania, the Transit category), showing selected sectors. There are three areas demonstrating a higher density of wind measurements, from the WNW (Ferrel, sector: 225-298°, width 73°), from the SE (Polar, sector 99-148°, width 49°) and from the S (Southerly, 148-217°, width 69°) . These sectors, capturing 76% of available wind measurements, are shaded in blue (Ferrel, 22%), grey (Southerly, 20%) and orange (Polar, 34%) respectively. (c): Frequency histogram of wind measurements plotted in 2b with 1° bin size. Ferrel (blue), Southerly (grey) and Polar (orange) sectors are highlighted.**

### 3.1 Categorisation by wind direction

Wind direction was the first variable used to categorise measurements as it is central to the definitions of the Ferrel and Polar cells (Aguado and Burt, 2015). Figure 2a shows a wind rose of all analysed measurements (February 6th to March 4th). Fig. 2b shows a wind rose for the period of the analysed measurements as in Fig. 2a, however excluding the ship's transit back to Hobart. The boundary between transit measurements and other categories was selected using wind speed and latitude. Measurements
taken north of 56.5° S were classified as transit and were associated with high wind speeds.
The return transit to Hobart was dominated by strong winds from the NW (evidenced by the high density of measurements from this direction plotted in Fig. 2a), as expected in the open Southern Ocean at latitudes between the seasonal ice zone and southern Tasmania. The aerosol characteristics of these open ocean measurements differed from those observed in measurements made within the seasonal ice
zone. Transit measurements make up 15% of the total measurements plotted in Fig. 2a.
In the restricted wind rose (Fig. 2b), there are three areas of increased measurement density, showing the prevalence of both WNW and SE winds during the voyage. These areas agree with those expected from the three-cell model of atmospheric circulation. The southern edge of the Ferrel cell is expected to display a dominant NW wind, and the northern edge of the Polar cell is expected to be associated with a

dominant SE wind (e.g. Constantin and Johnson, 2021, Sadler et al., 2011, Neiburger et al., 1971) .
There also exists frequent measurements with a southerly wind, possibly linked to katabatic outflow
from the Antarctic continent.

Measurements taken south of 56.5° S were then divided into one-degree wind direction bins and a
frequency histogram plotted, presented in Fig. 2c. The frequency of wind measurements in each bin was
used to create sectors of increased measurement density. Bins which contained greater than 0.2% of
total wind measurements encompassing the highest-density measurement regions were selected for
analysis. Three sectors were created, associated with WNW winds and labelled Ferrel (225-298°, 73°
width, n = 1444), SE  winds labelled Polar (99-148°, 49° width, n = 2211), along with an additional
Southerly sector (S, 148-217°, width 69°, n = 1275). These sectors, shaded in Fig. 2b and c, contain
22%, 34% and 20% of total wind measurements, respectively. Wind direction was variable during the
period of the voyage in which the ship was within the seasonal ice zone, indicating periods of influence
from both the Ferrel cell and Polar cells. Measurements from the southerly sector (148-217°) have been
classified separately due to the potential influence of  katabatic outflow on these measurements
(Chambers et al., 2018). These southerly data are assumed to have been influenced chiefly by
orographic factors and therefore any assumption regarding the air mass origin is difficult to make.
Analysis of aerosol measurements from the southerly sector suggest influence from both Polar and
Ferrel categories. Wind measurements not classified as Ferrel, Polar, Southerly or Transit were assigned
"Other". Boxplots of these five classifications are presented in Figure 4.

A sensitivity analysis was performed on the borders of the Ferrel and Polar sectors and the results of
this analysis are presented in the supplementary Table S1. Sectors were shifted 10° closer to 360°, 10°
closer to 0°, symmetrically widened by 20° and symmetrically narrowed by 20°. The median $CN_3$
concentration of the selected Ferrel measurements is higher than or equal to each of the shifted Ferrel
cell sectors, with all shifted sectors displaying median concentrations between 229 and 265 cm$^{-3}$. The
selected Ferrel sector median $CN_3$ concentration is 265 (CI 252-279) cm$^{-3}$. Adjusting the Polar cell
sector introduces more variability in the median $CN_3$ concentration: adjusted sector concentrations
range from 534-620 cm$^{-3}$. The median $CN_3$ concentration of the selected sector, 594 (CI 573-615) cm$^{-3}$,
is in the upper region of this range. Thus, adjusting selection boundaries is expected to cause only small
changes in the median particle concentration of the Ferrel and Polar cell sectors as a non-uniform
particle source was being measured from a moving platform. The relatively small changes observed
with shifted sectors demonstrates the median aerosol statistics reported are robust to boundary changes
in selected wind sectors.

### 3.2 Categorisation by absolute humidity

Using measurements from the same voyage as discussed here, Chambers et al. (2018) found unexpected
diurnal variability in radon concentrations during the voyage which was explained as arising from
katabatic outflow events. Events classed as katabatic outflow were associated with higher
concentrations of CCN. A mid-morning radon minimum was especially indicative of this type of
transport. Antarctic katabatic outflow air masses are often of free tropospheric origin (Chambers et al.,
2018, Nylen et al., 2004) and are expected to be drier than those that have extended residence times
over the local marine environment. Therefore, absolute humidity (AH) was the second meteorological

variable used for categorising aerosol measurements. Categorisations were performed on the same set of measurements used for division by wind direction. Each categorisation (whether by wind or AH) is therefore a subset of the total voyage measurement superset. Wind and AH categories overlap to differing degrees: a summary of overlaps is presented in Table S2.

Absolute humidity was calculated using a derivation of the ideal gas law (Carnotcycle, 2012) from
relative humidity (derived from wet- and dry-bulb temperature measurements) and measured ambient temperature using the following equation, where $AH$ is absolute humidity in units g m$^{-3}$, $T$ is measured temperature in degrees Celsius, $RH$ is relative humidity expressed as a percentage, $MW_{H_2O}$ is the molecular mass of water and R is the Ideal Gas Constant.

$$AH\ (g\ m^{-3}) = \frac{6.112e^{\frac{17.67T}{T+243.5}} \times RH \times MW_{H_2O}}{(273.15 + T) \times 100 \times R}$$

A frequency histogram, presented in Figure 3, was constructed from the timeseries of AH measurements. Examining the distribution allowed three AH categories to be defined. These are defined in the first instance by a local minimum at 4.30 g$_{H2O}$ m$^{-3}$ in the frequency histogram of AH
measurements. Measurements below this threshold were designated as Low AH. Measurements in the region 4.30-5.75 g$_{H2O}$ m$^{-3}$ were classified in the Mid AH category. Finally, measurements above 5.75 g$_{H2O}$ m$^{-3}$ were classed in the High AH category. 45% (n=3473) of valid AH measurements were classified as Low AH, 39% (n=3028) as Mid AH and 14% (n=1125) as High AH. The Low AH category is expected to capture air masses with either a katabatic flow or free-tropospheric source
region as air from these sources have not had an extended residence time in a marine environment. The Mid AH is expected to consist largely of marine air masses. Note that the High AH category overlaps almost exclusively (>99%) with the Transit category as defined above. This overlap is expected for maritime measurements as the water saturation pressure is dependent on temperature and ambient temperature increased during the transit from the Antarctic coast to Tasmania.

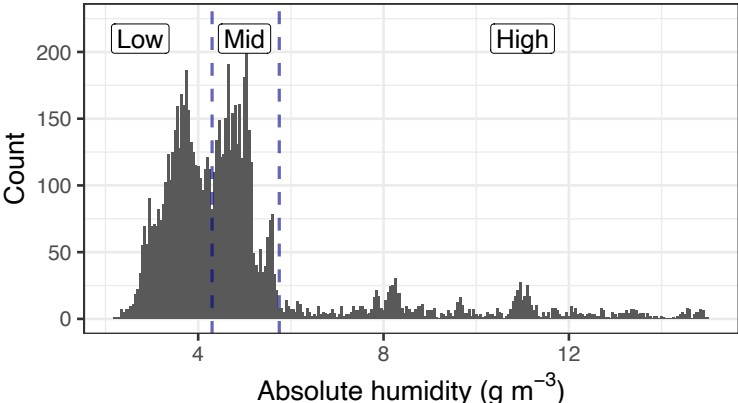

Figure 3: Frequency histogram of absolute humidity measurements including absolute humidity categories used for data analysis. The Low-Mid boundary is at 4.30 g$_{H2O}$ m$^{-3}$, and the Mid-High boundary at 5.75 g$_{H2O}$ m$^{-3}$.

### 3.3 General properties of observed aerosol populations

Measurements were made for the entire voyage, but as noted above an instrument malfunction meant $CN_3$ measurements are not available until February 6, 2017. For the period where all measurements are available the data have been summarised as boxplots (Fig. 4a, b and c present boxplots of $CN_3$, $CCN_{0.55}$ and $^{222}Rn$). The median $CCN_{0.55}$ concentration of 167 (CI 158-176) cm$^{-3}$, implies approximately half the particles measured as $CN_3$ could be activated as CCN (concentration of 354 (CI 344-363) cm$^{-3}$). The median $^{222}Rn$ concentration observed during the voyage was 64.5 (CI 62 - 67) mBq m$^{-3}$. The low concentrations of both aerosol number and $^{222}Rn$ suggests that measurements of clean marine air dominated, and that the air masses sampled are largely free of terrestrial influence. For comparison, the mean summer $CCN_{0.5}$ concentration from the Cape Grim 'baseline' from measurements taken in the 1990's was reported as 100 cm$^{-3}$ (Gras, 1995). A similar summer 'baseline' median concentration has been reported from the site more recently (Gras and Keywood, 2017). The Cape Grim 'baseline' sector is specifically selected as being free from terrestrial and anthropogenic influence and thus can be assumed to represent the remote Southern Ocean background signal. The median $^{222}Rn$ concentration observed in 'baseline' measurements (2001-2008) at Cape Grim was 42 mBq m$^{-3}$ (Zahorowski et al., 2013). In comparison, the non-baseline median for the same period was 372 mBq m$^{-3}$, nearly a factor of ten larger. The low concentration observed in the present study suggests sampled airmasses are free from recent terrestrial influence and are representative of Southern Ocean boundary layer conditions. Overall particle number concentration and CCN concentrations reported here are also similar to those reported by Schmale et al. (2019) as $CN_7$ (leg 1 median: 470 cm$^{-3}$) and $CCN_{0.2}$ (leg 1 median: 114 cm$^{-3}$) during the Antarctic Circumnavigation Expedition (ACE-SPACE), though the activation ratio observed during ACE-SCPACE was lower than that observed during PCAN.

A median size distribution plotted from all valid SMPS measurements for the analysis period during PCAN (supplementary Figure S1) has a bimodal shape indicative of cloud-processed remote marine aerosol populations (Hoppel et al., 1990), with the Hoppel minimum close to 70 nm. An Aitken mode is evident, centred at 38.5 nm. A less concentrated accumulation mode centred at 126 nm is also present. This distribution is similar to that observed in marine polar airmasses during a summer voyage close the Antarctic peninsula (Fossum et al., 2018), though smaller in magnitude. Previous aerosol number concentration measurements in the East Antarctic seasonal ice zone suggested the influence of large-scale atmospheric transport on locally observed aerosol concentrations (e.g. Humphries et al., 2021, Alroe et al., 2019). In these PCAN measurements, immediate changes in aerosol number concentration suggestive of an atmospheric boundary at the polar front, measured as $CN_3$ or $CCN_{0.55}$, were not observed to the extent of previous campaigns in the region (Alroe et al., 2019, Humphries et al., 2016). It must be considered at this point that the polar front describes a feature of climatological atmospheric circulation, which is influenced on a shorter timescale by meteorological conditions. Therefore the strength of this atmospheric boundary is expected to vary over the synoptic timescale. Meteorological variables have been used to place aerosol measurements into the framework of the Polar and Ferrel (mid-latitude) cells.

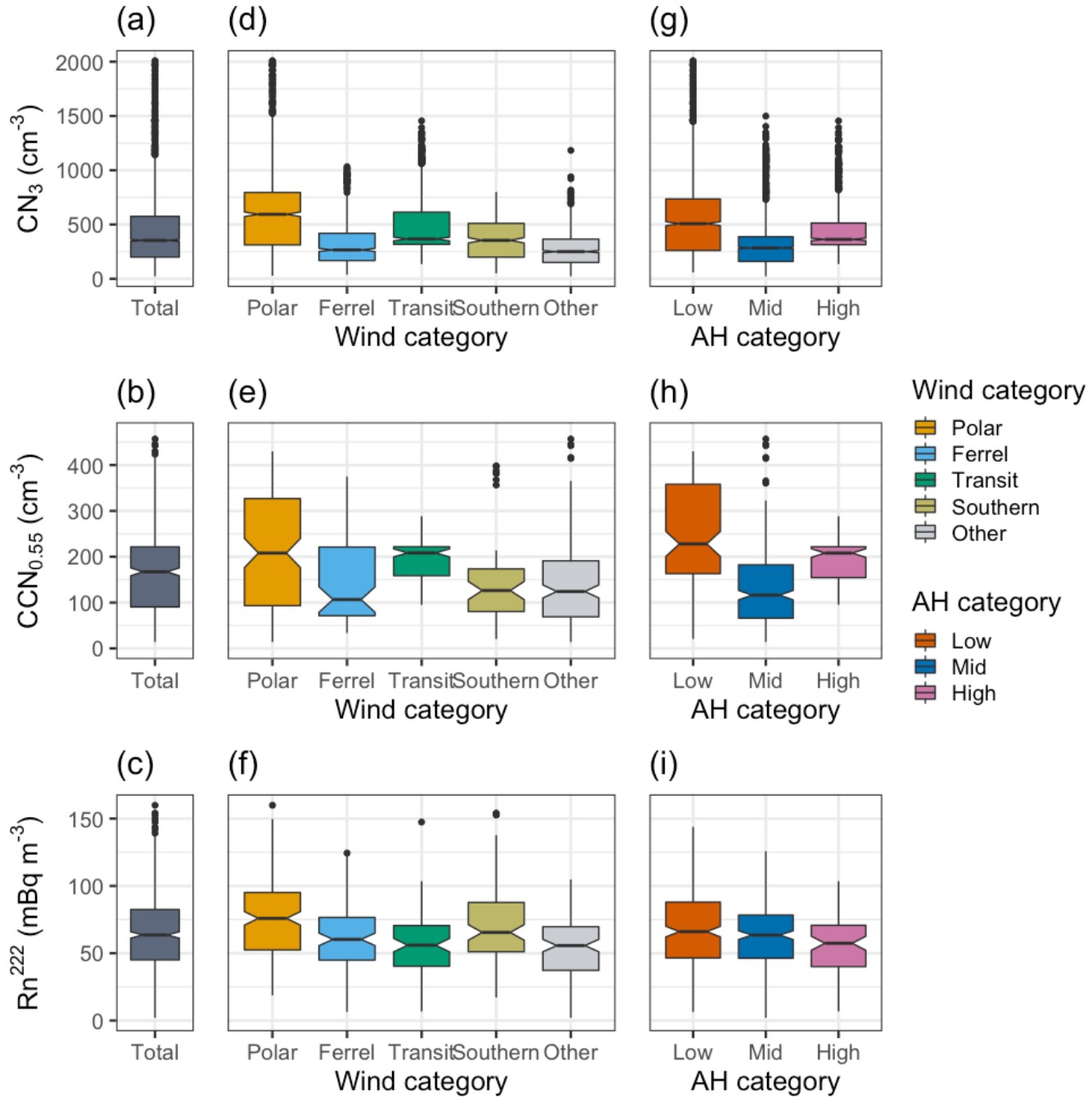

**Figure 4: Box plots for all analysed measurements and divided by category for (a), (d) and (g) $CN_3$; (b), (e) and (h) $CCN_{0.55}$; and (c), (f) and (i) $Rn^{222}$. Plots in the centre column are split by wind category and plots in the right-hand column are split by absolute humidity. Notches indicate the 95% confidence intervals in the median. Black dots represent outliers, defined as points greater than 1.5 times the interquartile ranges distant from the nearest quartile. Medians and confidence intervals for each variable in each category are reported in the supplementary Table S3.**

395

## 3.4 Aerosol variables and radon grouped by wind direction

Fig. 4 presents box plots of $CN_3$ (Fig. 4d), $CCN_{0.55}$ (Fig. 4e) and radon concentrations (Fig. 4f) grouped by wind direction. Statistics represented graphically in Fig. 4 are also presented in tabular form in supplementary Table S2. In each case the median concentration is higher in Polar cell measurements than in Ferrel cell measurements. Median $CN_3$ concentration in the Polar cell, 594 (CI 573-615) $cm^{-3}$ is approximately a factor of two larger than the Ferrel cell median, at 263 (CI 250-277) $cm^{-3}$. Median $CCN_{0.55}$ concentrations in the Polar cell were 208 (CI 176-240) $cm^{-3}$, compared to a median concentration of 113 (CI 84-142) $cm^{-3}$ in the Ferrel cell. Median radon concentration in the Polar cell was 78.3 (CI 73.3-83.3) mBq m$^{-3}$ compared to 59.1 (CI 54.3-63.9) mBq m$^{-3}$ in the Ferrel cell. For both aerosol variables the transit category median is between those observed in the Polar and Ferrel (Fig. 4). The higher median concentrations of $CN_3$ and $CCN_{0.55}$ observed in the Transit category (compared to the Ferrel) is likely due to the increased influence of continental and anthropogenic aerosol sources on boundary layer aerosol populations in the northern latitudes of the Southern Ocean, particularly above 50°S, as reported by Humphries et. al (2021). Measurements reported here as Ferrel cell were taken too far south to be significantly influenced by these additional particle sources.

In the Southern sector the median $CN_3$ concentration is 354 (CI 329-379) $cm^{-3}$, between those observed in the Polar and the Ferrel. As discussed in Section 3.1, Southern measurements are separated from the Polar/Ferrel classification in an attempt to isolate the known influence of katabatic outflow on near-coast measurements of boundary layer aerosol in the East Antarctic seasonal ice zone (Chambers at el., 2018). It is recognised that identifying katabatic flow by wind direction is not ideal. However, given the context of classifying measurements by wind direction, this method has been used in this study for consistency. The remaining 24% of measurements of uncertain air mass history, classified as 'Other', have a mean $CN_3$ concentration of 250 (CI 240-250) $cm^{-3}$. This is lowest median $CN_3$ concentration observed in any category. The lowest median $^{222}Rn$ concentration was also observed in the 'Other' classification, suggesting air masses sampled may have sources in the remote atmosphere distant from aerosol precursor sources and continental influence.

Median aerosol size distributions divided by wind sector are presented in Figure 5a. Each category displays a bimodal distribution. The Transit distribution is the most strongly bimodal. This is indicative of heavily cloud-processed marine aerosol (Hoppel et al., 1990) and results in one modal maximum in the Aitken mode (37.2 nm) and the other in the accumulation mode (157 nm). The median magnitude of the modes are 421 (CI 378-464) and 288 (CI 251-326) $cm^{-3}$ respectively (Fig. 5a).

In contrast, the distributions plotted for the Ferrel and Polar cells differ from the Transit period distribution. Both Polar and Ferrel distributions contain two local maxima with a smaller degree of bimodality. It should be noted that all aerosol populations, even those potentially descending from the free troposphere, had the potential to be cloud processed before sampling. Thus, the observed bimodality in all categories is not surprising. The Polar airmass has a higher particle concentration across the distribution than the Ferrel (as expected accounting for the larger median $CN_3$ concentration). The difference is relatively uniform across the distribution between 20 and 120 nm. The Polar and Ferrel distributions have modes in similar size ranges: the smaller mode in the Polar cell has a maximum at 38.5 nm, and at 40 nm in the Ferrel. The larger mode is at a slightly larger diameter in the

Ferrel cell dataset (109 nm) than the Polar cell (101 nm). Modal sizes and concentrations from Fig. 5 are summarised in the supplementary Table S3.

Higher particle concentrations in the Polar cell have been observed previously in the east Antarctic seasonal ice zone. In addition to the spring 2012 campaign SIPEXII (Humphries et al., 2016) and identification of katabatic influence on local CCN concentration (Chambers et al., 2018) referred to previously, there is evidence for a change in aerosol population across the polar front during a summer latitudinal transect of the Southern Ocean in 2016 (Alroe et al., 2019), along with variability in the Aitken mode dependant on marine biological precursors and synoptic scale systems. Accumulation mode peaks in both the Ferrel and Polar median size distributions are less concentrated than in the Aitken mode.

These particle concentration characteristics contrast with observations made by Schmale et al. (2019) in the Western Antarctic sector of the Southern Ocean. Using k-means clustering, Schmale et al. observed two clusters of aerosol size distribution clusters with different air mass histories. Cluster 1 was indicative of extended residence in the Southern Ocean marine boundary layer (similar to the Ferrel category identified here), while the other, cluster 2, showed influence of descent over the Antarctic continent and polynya regions. The modal sizes of cluster 1 distribution are larger than those observed in the present Ferrel measurements. The same is true when comparing the cluster 2 distribution to the present Polar distribution. This information, coupled with the more concentrated accumulation mode observed by Schmale et al., suggests the particle populations observed during PCAN may have different air mass histories than those observed in the West Antarctic. Evidence for differing aerosol regimes near the East and West Antarctic coasts are evident in the differences in observed particle concentrations and activation ratios during the 2016-2017 austral summer (Schmale et al., 2019).

To further characterize the observed size distributions, the modal sizes were determined. That is, the size bin with highest median concentration in the Aitken and accumulation size range is selected as the mode. To estimate the uncertainty in the modal particle size for Aitken and accumulation mode aerosol, the bin with maximum concentration was retrieved from each individual distribution and analysed. Details of this analysis are found in the supplementary information S1. These results, summarised in the supplementary Table S4, are different to the modal bin sizes retrieved for the overall median distributions noted above. This is a result of the distributions changing as concentrations change. While significant differences between the modal size bins do exist between categories when calculating the uncertainty using this method, these differences are small in magnitude (less than 12 nm in each mode between Polar and Ferrel, and between Low and Mid AH) for both wind and AH categorisations. It is therefore unclear what conclusions can be drawn from this result. The median mode of individual distributions are plotted with 95% CI are plotted for each category in Fig. 5.

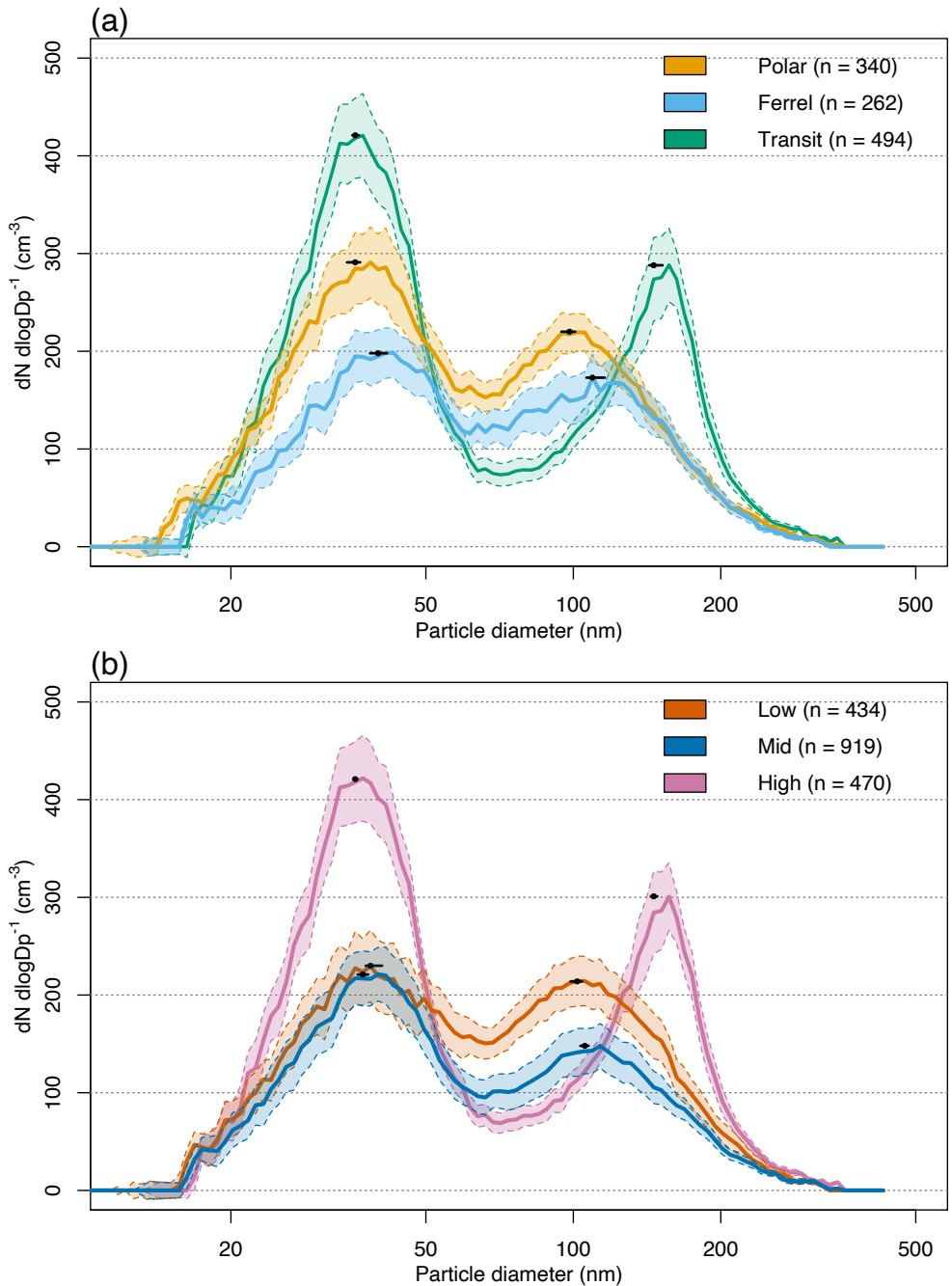

**Figure 5: Median particle size distributions for (a) each wind category and (b) each AH category. Shaded areas represent the 95% confidence interval in the median for each bin size. Note the diameter axis is logarithmic in scale. The modal sizes and concentrations for each distribution are reported in the supplementary Table S3. Black points plotted on each distribution indicates the median Aitken and accumulation mode generated from analysis of individual size distributions. Error bars represent at 95% confidence interval. More detail is provided in the supplementary information S1, and Table S4.**

## 3.5 Aerosol variables and radon grouped by absolute humidity

The categorization by absolute humidity should allow identification of the impact of katabatic outflow on the aerosol population. Fig. 4 displays boxplots of $CN_3$, $CCN_{0.55}$ and radon concentrations grouped by absolute humidity category. Measurements of $CN_3$ and $CCN_{0.55}$ in the Low AH category are higher than those in the Mid and High AH categories. Median $CN_3$ and $CCN_{0.55}$ concentrations in the Low AH category were 507 (CI 489-526) cm$^{-3}$ and 228 (CI 201-255) cm$^{-3}$ respectively. In comparison, the Mid AH medians for $CN_3$ and $CCN_{0.55}$ were 284 (CI 276-292) cm$^{-3}$ and 117 (CI 107-128) cm$^{-3}$, respectively. This is consistent with the finding that katabatic outflow events are associated with higher aerosol concentrations than locally sampled air masses observed by Chambers et al. (2018), as katabatic outflow events are likely included in the Low AH category due to the dry nature of katabatic airmasses. While not statistically significant, a difference in radon concentration is also noted between the categories: a median of 68.1 (CI 64.0-72.1) mBq m$^{-3}$ in during Low AH compared to 63.6 (CI 60.3-66.9) mBq m$^{-3}$ during Mid AH. It is likely that the Low AH category captures the katabatic outflow present in Type 3 and Type 4 days presented in Chambers et al. (2018). Type 4 days were defined by Chambers et al. using a mid-morning $Rn^{222}$ minimum, daytime winds from the S/SW and relatively strong winds, indicative of air masses strongly influenced by katabatic outflow. Type 3 days shared similar characteristics to Type 4 days though to a lesser extent. The median $CCN_{0.55}$ concentration observed in the present analysis for Low AH agrees well with maximum CCN concentrations reported for mid-morning katabatic outflow (Type 3 and 4 days) CCN peaks (Chambers et al., 2018). It should be noted that the Low AH category includes a greater percentage of the measurements (45%) compared to the Type 3 and 4 days reported by Chambers et al. (2018) (~27%) so likely includes measurements not influenced by katabatic drainage.

Median size distributions plotted for each AH category are presented in Fig. 5b. As noted earlier, the High AH category is also the Transit category, and hence a measurement of open ocean Ferrel cell aerosol. Similar to Fig. 5a, each distribution shows bimodal character. The Low AH and Mid AH have a smaller magnitude of bimodality as there is a smaller concentration difference between modes and the Hoppel minimum compared to the Transit distribution. The smaller mode in each of these distributions, in the Aitken range, is similar in magnitude and concentration: at 38.5 nm with concentrations of 230 (CI 193-266) cm$^{-3}$ in Low AH and at 40.0 nm, concentration 221 (CI 194-249) cm$^{-3}$ in Mid AH. The larger mode is more concentrated in the Low AH category, with a concentration of 214 (CI 189-240) cm$^{-3}$ at 106 nm, than the Mid AH: concentration 148 (CI 125-171) cm$^{-3}$ at 113 nm. The significance of the results presented above will be discussed in the following paragraphs.

## 3.6 Evidence for large-scale atmospheric transport influence on Southern Ocean aerosol populations

Differences in $CN_3$ and $CCN_{0.55}$ concentration are evident when meteorological variables are used to group aerosol measurements. Grouping measurements by wind direction produces higher median concentrations of $CN_3$ and CCN sampled during wind directions indicative of Polar cell air. Similarly,

higher concentrations of $CN_3$ and $CCN_{0.55}$ were observed when AH is less than 4.3 $g_{H2O}$ m$^{-3}$. The Southerly and Polar cell air is expected to be drier than Ferrel cell air (which has a theoretically longer residence time in the marine boundary layer as well as having increased contact with warmer ocean water). These results provide evidence that large-scale atmospheric transport (i.e. the atmospheric

transport included in the Polar and Ferrel cells) influences local aerosol populations in the east Antarctic seasonal ice zone. Note, however, that the aerosol size distributions sampled under different wind and AH regimes are similar. The aerosol size distributions are also similar between Polar and Ferrel wind measurements. The similarity between these median size distributions, as pictured in Fig. 5, suggests that a common aerosol source may contribute to these populations.

Back trajectory analysis can be used to further identify influence of large-scale atmospheric processes on boundary layer aerosol in the east Antarctic seasonal ice zone. HYSPLIT back trajectories were run for each hour, on the hour, of the voyage. The ship's location at each time was used as the trajectory end point. Trajectories were separated into wind and absolute humidity categories using a similar method to that used for the aerosol measurements. The classification category for the wind direction or AH at the

trajectory endpoint was used to classify the trajectory. Median vertical profiles, frequency histograms and trajectory maps for each category were constructed, presented in the supplementary Figures S2 and S3.

Distinct air mass histories are expected to contribute to different aerosol population characteristics observed in each classification. Back trajectory analysis suggests a greater free tropospheric influence is

present in the Polar wind category compared to the Ferrel wind category. The boundary layer height in this region of the Southern Ocean has been measured to be 900±400 m north of the polar front and 700±200 m south of the polar front during summer and early autumn voyages (Alexander and Protat, 2019). The median profile for the Polar category shows air masses residing above 1000 m above ground level 48 hours before the trajectory endpoint. Sampled air masses in the Polar category are therefore

likely to have significant free-tropospheric influence. The median profiles for the Ferrel and Transit wind categories show no such development, instead keeping close to the surface for the duration of the 72 hours modelled (Fig. S2). The large interquartile ranges on this plot reflect significant variability in airmass altitude. Interestingly, the trajectories for the Southerly category also show history of descending airmasses.

A similar result is generated when examining trajectories grouped by AH category. The Mid and High AH category median trajectory vertical profiles show little vertical development in the 72 hours prior to measurement (Fig. S3). In contrast, the Low AH category median vertical profile demonstrates vertical development comparable to that of the Polar wind category with significant residence time at free tropospheric altitudes in the 72 hours prior to sampling.

Back trajectories indicate a greater free tropospheric influence on airmasses sampled under Polar wind regimes and with low AH than during other measurement periods. Higher particle counts ($CN_3$ and $CCN_{0.55}$) are also observed in these air masses. Observation of higher aerosol number concentrations and a free tropospheric influence on air masses sampled in the Polar cell in the east Antarctic seasonal ice zone are common to the summer measurements presented here and the spring 2012 measurements

reported in Humphries et al. (2016). However, there are some key differences between measurements suggesting different source and air mass transport mechanisms.

**3.7 Evidence for seasonality in seasonal ice zone regional aerosol sources**

Comparing SIPEXII Polar cell aerosol measurements from spring 2012 to the PCAN Polar measurements (classified by wind direction) indicates there may be different sources, or source
strengths, for populations between spring and summer despite some similar characteristics. Humphries et al. (2016) defined Polar cell measurements using a clear change in observed aerosol population along with detailed back trajectory analysis. This method is different to that used here. However the persistent nature of dominant surface winds associated with Polar and Ferrel cells, coupled with back trajectories (plotted in supplementary Fig.S2 and Fig S3) aligning with those modelled for SIPEXII measurements
allows comparison between Polar cell measurements to be made between the voyages. The median $CN_3$ concentration in the Polar cell during SIPEXII was 816 cm$^{-3}$ (Humphries et al., 2016), compared to a median Polar $CN_3$ concentration of 594 cm$^{-3}$ reported here. The median SIPEXII Ferrel cell concentration was 196 cm$^{-3}$, compared to a median of 263 cm$^{-3}$ during PCAN. The magnitude of the change between the Polar and Ferrel cell during SIPEXII was a factor of approximately four whereas
for PCAN this change is closer to a factor of two. It is important to note that these measurements were taken from different platforms, and therefore different particle inlet transmission efficiencies must be considered. However, the difference in the magnitude of enhancement observed in the Polar cell, which does not depend on inlet transmission efficiency in both voyages demonstrates that there is a robust difference in aerosol populations observed between voyages.
A further difference between spring and summer measurements becomes evident when considering the size of particle populations measured during the voyages. The Ferrel-Polar cell transition during SIPEXII was especially prominent in the $CN_{3-10}$ fraction, with the median $CN_{3-10}$ concentration in Ferrel cell reported at 45 cm$^{-3}$ compared to a median of 776 cm$^{-3}$ in the Polar cell. This is attributed to a change of aerosol source by Humphries et al. (2016). A lack of a $CN_{10}$ measurement on board during
PCAN (due to instrument malfunctions) prevents a direct comparison of this metric between voyages. Aerosol size distribution measurements from PCAN can be used however: the SMPS instrument used on this voyage observed measurements as small as 8 nm diameter during the voyage, allowing the particle concentration to be calculated in the $CN_{3-8}$ range. During PCAN, the median $CN_{3-8}$ concentration in the Ferrel cell was 118 cm$^{-3}$, compared to 148 cm$^{-3}$ in the Polar cell. This suggests the
difference in aerosol number concentration observed during PCAN is not driven by a large change in concentration of very small particles as observed during SIPEXII.
Polar cell measurements from SIPEXII and PCAN (defined by wind direction) were compared to investigate this further. Figure 6 displays four correlation plots: AH measurements and atmospheric pressure measurements from PCAN (top row) and SIPEXII (bottom row) are plotted against $CN_3$
measurements. Events of anomalously high $CN_3$ concentration during SIPEXII as identified by Humphries et al. (2016) are excluded from this analysis. $CN_3$ shows a negative relationship with absolute humidity in PCAN Polar cell measurements ($r = -0.50$). This relationship is weaker for SIPEXII measurements ($r = -0.26$). In contrast, a negative relationship is observed between SIPEXII Polar cell $CN_3$ measurements and atmospheric pressure ($r = -0.61$), discussed in detail in Humphries et
al. (2016). This relationship is weaker in the PCAN Polar cell measurements ($r = -0.23$). This suggests the source regions or mechanisms of measured Polar cell aerosol in the east Antarctic seasonal ice zone may vary between seasons. A free tropospheric injection model presented by Humphries et al. (2016),

assosciated with events of local low atmospheric pressure, was used to account for unexpectedly high concentrations of small aerosol during SIPEXII. This tropospheric injection may not be the dominant source of Polar cell aerosol measured during summer 2017. This conclusion was also drawn from the analysis of radon diurnal cycles and CCN concentrations from the PCAN voyage (Chambers et al., 2018), which highlighted the contribution of katabatic outflow to the aerosol populations of the near-shore atmosphere of the east Antarctic.

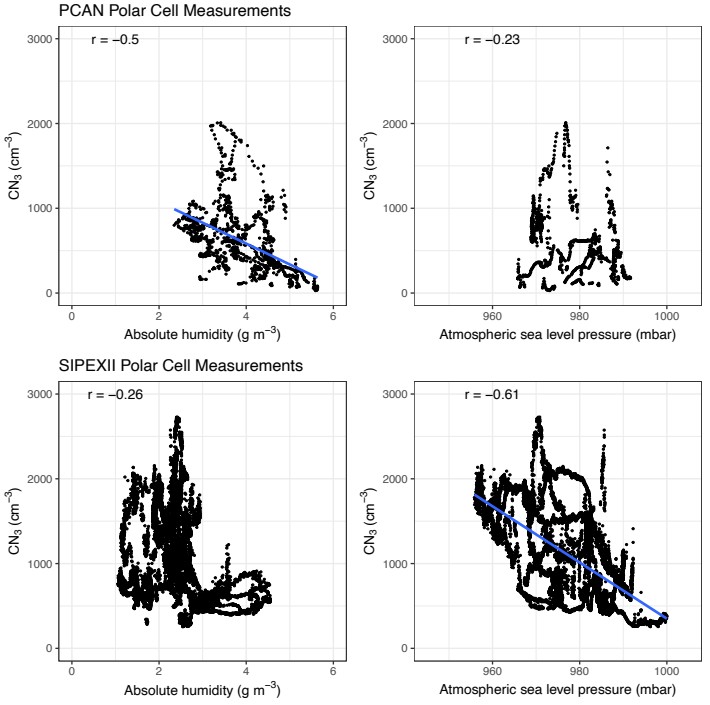

**Figure 6: Comparison of PCAN (top row) and SIPEXII (bottom row) AH and atmospheric sea level pressure measurements plotted against $CN_3$ measurements in the Polar cell. Linear regression lines are plotted on the stronger relationship for each voyage. Pearson's $r$ values are plotted on each facet.**

## 3.8 Explaining the observed seasonality

Higher concentrations of aerosol number concentration (expressed as $CN_3$, $CN_{10}$ or CCN) are observed in the Polar cell influenced airmasses in the east Antarctic seasonal ice zone during both spring and summer compared with Ferrel cell airmasses. However, the magnitude of this change is different between seasons. Though a direct comparison cannot be made, size-resolved measurements suggest different size distributions between voyages. Therefore, it seems different source and transport mechanisms may influence Polar cell aerosol populations in the seasonal ice zone in spring and summer.

During spring, the circumpolar trough (an alternative name for the polar front), the region of low pressure surrounding the Antarctic continent, is at peak intensity and located closest to the Antarctic

coast (Pook, 2002). This serves to strengthen westerly winds in the southern Ferrel cell and intensify the pressure gradient across the trough. This may be associated with limited atmospheric transport across

the polar front. This could cause the Polar and Ferrel cells to be more isolated from each other, creating a 'boundary' observable in aerosol populations. Stronger low-pressure systems are present in the region in spring (evidenced by the lower atmospheric pressures measured during SIPEXII compared to PCAN, Fig. 6) compared to summer. These strong low-pressure events inject $CN_3$ into the Polar cell boundary layer from the free troposphere (Humphries et al., 2016) causing the correlation between atmospheric

pressure and $CN_3$ concentration observed in the SIPEXII measurements.

The circumpolar trough weakens in summer (Pook, 2002). This may allow more inter-cell atmospheric transport between the Polar and Ferrel cells. A lack of clear 'boundary' observable in aerosol populations in PCAN measurements between the cells could be symptomatic of this, as are similar median aerosol size distributions for measurements classified as Polar cell and Ferrel cell. Low pressure

systems in the region are generally weaker in summer, and therefore the free tropospheric injection of small particles could be less dominant in the Polar cell. Instead, increased influence of the katabatic outflow on measurements in the Polar cell documented by Chambers et al. (2018) is observed and strengthens the relationship between $CN_3$ concentration and AH. It is recognised that katabatic outflow from the Antarctic continent has an annual minimum in summer (Parish and Cassano, 2003). However,

during periods of limited insolation, downslope drainage driven by latent cooling still occurs during the summer months, especially in the high-wind East Antarctic (e.g. Yu et al., 2020, Chambers et al., 2018). It must also be considered that differences in tropospheric radon at coastal Antarctic stations between summer and winter provide evidence for seasonality in the impact of recent terrestrial influence on the remote Southern Ocean boundary layer (Chambers et al., 2018).

It is clearly possible, however, to have periods in any season with strong low-pressure systems and so this hypothesis requires testing over varying meteorological conditions. Other mechanisms of air mass exchange between the boundary layer and free troposphere also exist, including boundary layer turbulence (e.g. Zhang et al., 2018, Driedonks and Tennekes, 1984). The extent to which the entrainment of free tropospheric air into the marine boundary layer driven by air mass exchange on a

scale smaller than the synoptic is also a question requiring more extended measurements in the region to answer. Results from a more recent summer voyage do observe distinct changes in aerosol properties when crossing the polar front (Humphries et al., 2021), demonstrating the complexity of the region.

A further difference between the seasons is the strength of aerosol sources. Phytoplankton blooms are known to occur around Antarctica in late spring and summer which can act as an additional particle

source via the production of aerosol precursor gases (Charlson et al., 1987). Increased organic fractions in summer Antarctic aerosol observations provide evidence for this mechanism (Hong et al., 2020). Concentrations of biogenic aerosol precursors are particularly high during periods of sea ice melt in the spring (Yan et al., 2020; Gabric et al., 2018) which may contribute to increased SOA formation during this period. Spring measurements taken during SIPEXII voyage were taken while significant

concentrations of sea ice surrounded the measurement platform, as indicated in the shading in Fig. 1. Very limited sea ice was present in these latitudes during the PCAN voyage. It is important therefore to consider the implications of ice-melt related algal bloom when comparing spring and summer measurements in the seasonal ice zone. The magnitude of the algal bloom is greatest in the waters off the west Antarctic Peninsula, but does still occur in the East Antarctic (Deppeler and Davidson, 2017).

The influence of phytoplankton blooms on ultrafine aerosol has been observed in this region before. Alroe et al. (2019) measured enhancements in Aitken and nucleation mode aerosol in the Southern Ocean off east Antarctica associated with air mass back trajectories passing over biologically productive regions. The annual cycle of this particle precursor source must therefore be considered when examining aerosol populations of the region.

**5 Conclusions**

Measurements of aerosol number concentration and aerosol size distribution have been reported here from the infrequently measured seasonal ice zone of East Antarctica. Total particle concentrations as $CN_3$ observed during the present summer voyage are similar to those observed elsewhere around the Antarctic continent. Cloud condensation nuclei concentrations are also similar here to those observed
previously, both from ship-based measurement campaigns and terrestrial stations situated at more northern latitudes. Particle size distributions indicated the majority of measurements were influenced by cloud processing to some extent.

This work provides more evidence for the growing body of literature suggesting that large-scale atmospheric transport processes influence the aerosol concentrations observed in the boundary layer of
the East Antarctic seasonal ice zone. Local winds from the SE, indicative of Polar cell air masses, displayed median $CN_3$ and $CCN_{0.55}$ concentrations much greater than when local wind direction was from the NW (indicative of Ferrel cell air masses). Similarly, aerosol measurements taken during periods where absolute humidity was less than 4.3 $g_{H2O}m^{-3}$ were also associated with enhancements in $CN_3$ and CCN number concentration compared to other periods of the voyage. Unlike previous voyages
to this region, obvious changes in aerosol concentration were not observed when crossing the atmospheric boundary at the polar front.

Evidence of seasonal differences of aerosol measurements in the Polar cell also emerges when comparing summer to spring measurements. Differing relationships between $CN_3$, atmospheric pressure and absolute humidity indicate that sampled aerosol populations may have modified source regions or
mechanisms.

Seasonal differences in aerosol populations observed in the East Antarctic seasonal ice zone may be explained using the following conceptual model. The distinct boundary in atmospheric composition observed during spring measurements may be a result of a strengthened circumpolar trough, isolating the airmasses of the Ferrel and Polar Cells. Polar cell aerosol populations are driven by tropospheric
injection of ultrafine particles into the boundary later. In contrast, the summer in this region generally exhibits a weakened circumpolar trough. This weakening would allow a greater air mass transfer between the Ferrel and Polar cells, and weaken the atmospheric boundary defined by sampled aerosol population. The reduction in strength of the Ferrel cell-Polar cell atmospheric boundary could reduce the difference between the two airmasses' aerosol populations. Synoptic low-pressure systems could be
expected to occasionally inject ultrafine particles from the free troposphere into the marine boundary layer. This may occur more frequently in spring due to more frequent, stronger synoptic systems. This case is not observed in the present measurements. Instead, katabatic outflow from the Antarctic continent more strongly influences particle number concentration enhancement.

This work provides evidence for a variability of tropospheric aerosol populations between seasons in
the east Antarctic seasonal ice zone. Measurements of air mass movements on the meso scale such as a
wind profiling boundary layer radar would also be of use in seeking to confirm source regions of
sampled aerosol populations and distinguishing air mass movements on a scale finer than that detected
by models. Greater understanding could be gained from measurement campaigns with durations longer
than the synoptic timescale as the discussed process have influence on a regional, rather than local,
spatial scale. Increased density of aerosol observations throughout the year, including additional
campaigns in summer and spring would also assist in understanding the complex factors driving aerosol
populations in the troposphere of the East Antarctic seasonal ice zone and in other locations surrounding
the Antarctic coast.

## 6 Code and data Availability

Software code in the language 'R' used in generating the present manuscript is available from the
corresponding author upon request. All data and samples acquired on the voyage are made publicly
available in accordance with MNF Policy. Measurements are available through the CSIRO Data Access
Portal at the following link: https://doi.org/10.25919/xs0b-an24.

## 7 Author contribution

RH instigated the PCAN investigation, along with JW and MK, and coordinated deployment. JS was
responsible for on-board data collection, data analysis and manuscript preparation. RH and SRW
provided scientific advice, data analysis expertise and manuscript preparation guidance. SC, AG and
AW installed and maintain the radon monitor on board the *RV Investigator*. IR was responsible for the
installation of many aerosol instruments on the *Investigator*. SG ran the offline HYSPLIT model. All
authors contributed to reviews of the manuscript.

## 8 Competing interests

The authors declare that they have no conflict of interest.

## 9 Acknowledgements

The authors gratefully acknowledge Dr. Leanne Armand, Chief Scientist on board voyage IN2017-V01,
and other voyage staff for allowing the PCAN piggyback project on board the *RV Investigator*. The
Authors wish to thank the CSIRO Marine National Facility (MNF) for its support in the form of sea
time on *RV Investigator*, support personnel, scientific equipment and data management with particular
thanks to the Seagoing Instrumentation Team. In particular we thank the technical and IT support
personnel on board the voyage. All data and samples acquired on the voyage are made publicly
available in accordance with MNF Policy. This project received grant funding from the Australian
Government as part of the Antarctic Science Collaboration Initiative program. The Australian Antarctic
Program Partnership is led by the University of Tasmania, and includes the Australian Antarctic

Division, CSIRO Oceans and Atmosphere, Geoscience Australia, the Bureau of Meteorology, the
Tasmanian State Government and Australia's Integrated Marine Observing System.

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
