# Peer review of "Summer aerosol measurements over the East Antarctic seasonal ice zone."

_Atmospheric Chemistry and Physics, 2020_

## Referee Comment (RC1) · Anonymous Referee #1 · 21 Dec 2020

The manuscript focuses on constraining aerosol measurements made in the East Antarctic seasonal ice zone using measured absolute humidity and wind direction. Using these meteorological variables to categorize measurements enable the authors to identify differences in measurements between air originating in the Polar and Ferrel cells and also identify air originating over the southern ocean or free troposphere/Antarctic continent. The measurements and discussion provided provide important context for understanding aerosol processes in this region of the Southern Ocean, where little is known. I believe the manuscript merits publication after considering the comments below. Over all, comments are minor and involve suggestions on manuscript organization and a couple small suggestions for added analysis.

Main comments:

[Figure]

Figure 1: Why are there no observations in one of the north/south transects of the SO? Ship exhaust? Or is it the software malfunction?

The sectors of wind direction focused on in the results accounted for 52% of the measurements. The results from this comparison are interesting, however some results on the other ~half of the measurements would be nice to provide a more complete picture, even if the air mass origin is not as precisely known. Excluding half of your measurements is a lot. The variability of aerosol properties from the southern sector are still important to know.

Why might transit aerosol (CN3, CCN0.55, distribution modes) be higher than the Ferrel cell category if the transit is largely in the Ferrel Cell?

Line 298-309 It is a little hard to follow this paragraph, but more importantly, it's unclear what the point of this paragraph is. If you are going to point out differences in mode diameters, some reasoning as to why they are different is necessary to justify the importance of identifying them. I am not sure you can provide some context for some of these small differences between mode diameters, but without it, the figure alone is enough to show the differences.

Line 343-352 Similar to the previous comment, you are essentially describing what is seen in the figures, but not providing context on why these differences are important or what they mean.

Line 318 – 339 Much of this comparison to Chambers et al. 2018 requires the reader to read Chambers et al. 2018 to understand it. It would be helpful to briefly include the necessary details from Chambers et al. 2018 to understand the comparisons you have made so that the reader does not have to read another manuscript to understand yours.

Section 3 is organized as if sections 3.1-3.4 are results and section 3.5-3.7 is the discussion. If you intend to keep the results and discussion separate, I suggest labeling

the sections as such (I.e. change section 3 to results and add a section 4 titled Discussion which would included section 3.5-3.7). Otherwise you leave the reader wondering where the discussion is (as I have shown in my previous comments).

I suggest moving Figures S2 and S3 to the main text as they provide important context.

Line 421. While this statement may hold true, comparing the tail of the size distributions is not the same as calculating the value of CN3-10. While you do not have CN10, you could still calculate CN3-8, which is a similar comparison.

Minor comments/suggestions:

The manuscript formatting does not follow the ACP requirements.

Line 56. Change 'far' to 'remote'

Line 121. add space between 'L min-1'

Figure 2. Maybe define the Ferrel cell wind direction as WNW to be consistent with the text. The width of Ferrel cell sector is 73 deg, not 68 deg.

Line 202. What are 'these' measurements

Line 207 15% of figure 2a or 2b?

Line 211. I know you defined the bin width elsewhere, but it may be worth restating here.

Line 212. Stick with 'sectors' and remove 'windows'

Line 223. The measurements are labeled "Southern" in Figure S1 (not "No Category")

Line 282 Reference the relevant figure

Line 292 What finding?

Line 299 Reported where?

Line 334 It would be better to explain what a 'Type 3 and Type 4' day is in Chambers et al. (2018)

Figure 6. change atmospheric pressure to atmospheric sea level pressure

Section 5 – Please provide a direct link to the data.

---

## Referee Comment (RC2) · Anonymous Referee #2 · 28 Dec 2020

Review of "Summer aerosol measurements over the East Antarctic seasonal ice zone" by Simmonds et al.

This study presents aerosol measurements from the PCAN field study performed on board the RV Investigator during summer 2017 in the seasonal ice zone of Antarctica south of Australia. Particle number concentrations with a diameter larger than 3 nm, as well as particle size distributions in the range from 8 to 507 nm and cloud condensation nuclei number concentrations at a supersaturation of 0.55 % were observed. The main objective of the work is to compare the summertime data against springtime data from a previous campaing, SIPEX-II, published by Humphries et al. (2016), where a marked step-change in concentration was found between the Ferrel and Polar cells. Here, the transition between cells is much smoother and data were hence separated

by wind direction and absolute humidity to infer differences. The authors find still larger concentrations in the Polar cell and speculate on reasons why that is and why there might be a difference towards springtime.

Overall, the measurements are well described and the analysis is straight forward. However, there are a number of points to be improved: The particle losses in the inlet are not described. It is also not clear whether the same inlet system was used in Humphries et al., which is important to know because this work is based on an intercomparison. In addition, the scientific motivation for this investigation is not well presented. It seems that the mere purpose of this study is to compare against Humphries et al. (2016), which is not much scientific merit per se. Better arguments for the relevance of this study need to be provided to make it publishable. Much more in depth discussion on the significance of the findings is needed. After reading the manuscript the question "so what?" is not answered. In addition, the data were analyzed selectively following a too narrow scientific scope, namely those data that fit the scheme of Ferrel and Polar cell. All data should be presented and discussed in the main text to provide a more comprehensive idea of aerosol in the so scarcely measured Southern Ocean. Below follow some comments, which might help the authors to strengthen their manuscript. I recommend a major revision before publication is considered.

Specific comments:

Abstract: I recommend rewriting the abstract. It should contain information on what was measured and when, on which platform etc. In the current form, it builds too much on the previous study by Humphries et al. and readers who are not familiar with that study will not get a good insight into what was done in this study. The abstract also heavily relies on knowledge of the Polar and Ferrel cells are, but those are not introduced. I recommend explaining what they are. It is also not clear whether the absolute humidity was measured or obtained from numerical weather prediction models. Furthermore abbreviations are not introduced.

Introduction

l. 49: It is unclear which model parameters the authors refer to.

l. 55: The argument why understanding the aerosol population over the East Antarctic seasonal ice zone is not very well fleshed out. Only a study of Shindell et al. (2013) is cited which highlights discrepancies between modeling and satellite observations. From this it is however not clear why this should researched further. This argument needs to be elaborated. I also recommend referring to recent publications on the topic, e.g. by Mc Coy et al. (2020) 10.1073/pnas.1922502117

l. 60-64: The discussion of results obtained from previous measurements over the Southern Ocean is extremely short and it is unclear what the main point is that the authors would like to make. Only two studies are cited and only one short result of one of the two studies is mentioned. Here, clearly, a more thorough review needs to be done and the main findings of several previous studies relevant to this particular work need to be synthesized.

l. 75 ff: Say where the cells roughly meet in terms of latitude.

At the end of the introduction, I expect a brief description of the campaign, i.e. when did it happen, what was measured...

Methods

Fig. 1: I recommend plotting the sea ice extent for SIPEX-II an PCAN since this is an important environmental factor for this study.

L. 110 ff: Meteorological measurements on ships can be biased because of the ship's superstructure (see e.g., Landwehr et al. (2020), 10.5194/amt-13-3487-2020). Did the authors do anything to check for potential biases and were the data corrected? A more detailed description is needed.

l. 115: Particles as small as 3 nm can easily get lost through diffusion. There is no

reason not to characterize the inlet losses at least theoretically, if experimentally this is not possible. The authors can use for example the particle loss calculator by Von der Weiden et al. (2009), www.atmos-meas-tech.net/2/479/2009/. Also the length of the inlet line is not given and neither are the bends described. This needs to added to the paper. The concentrations reported in this work might be used for satellite and model validation, hence providing uncertainties with the numbers, and those include inlet losses, is essential.

l. 120: How were size distributions measured? One expects this information here, where they are mentioned not further below.

l. 140: There is no mention of what Rn measurements were used for and why.

l. 145 ff: How were $CO_2$ and black carbon measured? This information needs to be included in the paper.

l. 160: Say which variables were used along the trajectories and which uncertainties are referred to? Do the authors mean the 3 D coordinates or the uncertainties in meteorological parameters?

l. 169: Explain what the McGill et al. (1978) style is. Most readers will not be familiar with it and will have to look up the reference.

Results

l. 175: The question is why one should expect a step change? Just because this was observed previously, once only, it does not mean that this is the norm. The authors should explain why their implicit assumption is the step change, or they should drop it and describe more neutrally that a step change has been observed previously, but one cannot expect it to be there permanently as it will depend on the meteorological situation. The Polar and Ferrel cells are descriptions of longer-term average circulation features. Those can deviate significantly on the shorter term. It is also unclear why it is so important to determine the transition between the two cells in the first place, and

in the second place why is it important to use aerosol parameters? Those two points are not very well motivated.

Fig. 2: All fonts are way too small. There are also substantial measurements near 180 ° in Fig. 2b. It is unclear why those are not considered in the analysis if the argument, as given in the caption, is the frequency of observations. Fig. 2c has no y-axis scaling.

l. 220ff: Why did the authors exclude measurements from the South? Those are also important and interesting. Just because they do not fit the scheme of Polar and Ferrel cell does not provide a good argument. I recommend data are shown and discussed. Also statistics for "no category" should be shown for completeness.

l. 241: Please provide the formula in the manuscript.

Fig. 3: The writing is too small.

l. 290ff: It is an interesting result that the accumulation mode diameter in the Polar cell is smaller than that in the Ferrel cell. This is opposite of what Schmale et al. (2019) found for the Western site of Antarctica. The authors should discuss their findings in a broader context, going beyond what has been found South of Australia. The question of larger particle number concentrations closer to Antarctica is not limited to that sector of the Southern Ocean. The author very briefly hint at potential causes in l. 485, but do not make the connection as suggested here.

Fig. 5: Diameter axes are typically logarithmic.

l. 334: the Types in Chambers et al. (2018) need to be explained here, otherwise it is not clear what the authors try to convey.

l. 333: Was "SO" introduced?

l.370ff: This is highly speculative. What are the observations that allow the authors to draw the conclusion that the lifetime of aerosols is longer in the Ferrel than the Polar cell? Particle lifetime is strongly determined by precipitation for the considered

size range. What are the differences between the cells? How long would it take to increase the size of an accumulation mode particle if the authors assumed it is sulfuric acid with a given accommodation coefficient? How long does coagulation take? Is this in line with the expected longer lifetime? Also, if the source is similar in both cells, why are there less particles in the Ferrel cell? The most likely explanation is precipitation scavenging, this would however counteract the lifetime argument. This statement clearly needs more elaborate discussion to back up the assumption.

l. 390: What are the vertical lines in figures S2b and S3b?

Conclusion:

l. 515: Why does it need strong low pressure systems to exchange air masses between the free troposphere and the marine boundary layer? Smaller convective processes during cloud formation and dissipation will do the same. Here more discussion is needed on how air masses might be exchanged.

l. 520: This work does not provide evidence for a seasonal cycle. It merely compares summer to spring measurements. There is no information on the fall and winter yet.

Technical remarks:

l. 128: Cloud condensation nuclei number...

---

## Author Comment (AC1) · 30 Mar 2021

Anonymous Referee #1 Received and published: 21 December 2020

The authors thank referee 1 for their time spent reviewing this manuscript and their valuable advice in suggested improvements. We have edited the manuscript in line with the suggestions from referee 1 as detailed below.

Figure 1: Why are there no observations in one of the north/south transects of the SO? Ship exhaust? Or is it the software malfunction? Yes, referee 1 is correct: measurements were not taken on one transect due to software malfunction. This has been clarified in the figure caption.

**... results on the other half of the measurements would be nice to provide a more complete picture, even if the air mass origin is not as precisely known...**

Thank you for this suggestion. Total-voyage plots for  $CN_3$ ,  $CCN_{0.55}$  and 222Rn have been included in Figure 4, along with the 'Southern' and 'No category' wind classifications. A total voyage median size distribution can now be found in the supplementary. I have also added a paragraph describing the overall voyage observations at the start of the results section. We agree that this adds useful depth to the manuscript.

**Why might transit aerosol (CN3, CCN0.55, distribution modes) be higher than the Ferrel cell category if the transit is largely in the Ferrel Cell?**

Thank you for this comment. The transit category of measurements represents a 'classic' remote marine bimodal aerosol distribution and encompasses the transit of the Investigator back to Hobart, whereas the 'Ferrel' category is representative of the southern extent of this regime where greater influences from, for example, biological secondary aerosols is expected. It is also likely that measurements from the Transit period of the voyage are influenced by continental and anthropogenic aerosol sources at their northern extent. In the 'Ferrel' category, these sources have little to no influence on local aerosol population as measurements are taken below 56°S. It must also be mentioned that homogenous aerosol concentrations in such large-scale atmospheric circulations are not expected- rather local sources and sinks will have a greater impact. I have clarified this and added an additional reference.

Line 298-309 It is a little hard to follow this paragraph, but more importantly, it's unclear what the point of this paragraph is. If you are going to point out differences in mode diameters, some reasoning as to why they are different is necessary to justify the importance of identifying them. I am not sure you can provide some context for some of these small differences between mode diameters, but without it, the figure alone is enough to show the differences.

Line 343-352 Similar to the previous comment, you are essentially describing what is seen in the figures, but not providing context on why these differences are important or what they mean.

Upon reflection, we agree this paragraph is difficult to understand and adds little useful detail to the manuscript. There it has been removed, (along with the similar paragraph at lines 343-352) and replaced with the following text.

"Significant differences between the modal size bins do exist between categories when calculating the uncertainty using this method. However, these differences are small (less than 12 nm in each case) for both wind and AH categorisations. It is therefore difficult to draw any conclusions from this result."

Line 318 – 339 Much of this comparison to Chambers et al. 2018 requires the reader to read Chambers et al. 2018 to understand it. It would be helpful to briefly include the necessary details from Chambers et al. 2018 to understand the comparisons you have made so that the reader does not have to read another manuscript to understand yours. Thank you for making this section more comprehensible. More details have been added to this section in an attempt to clarify how the present work compares to Chambers et al. (2018) in order to improve the clarity of the manuscript.

Section 3 is organized as if sections 3.1-3.4 are results and section 3.5-3.7 is the discussion. If you intend to keep the results and discussion separate, I suggest labelling the sections as such (I.e. change section 3 to results and add a section 4 titled Discussion which would included section 3.5-3.7). Otherwise you leave the reader wondering where the discussion is (as I have shown in my previous comments).- Thank you for this suggestion. The authors believe sections 3.5-3.7 contain most of the discussion, though some results are still being presented in these sections. Therefore, I have decided to leave the structure organised as it is, however, have included a clarifying sentence at the end of Section 3.4 ("The significance and implications of the results presented above will be discussed in the following paragraphs along with other supporting evidence.")

*I suggest moving Figures S2 and S3 to the main text as they provide important context.* – Thank you for this suggestion. The authors are not sure this is the best approach- these figures have been included in the main text in previous versions, but on the advice on multiple co-authors have moved them to the supplementary to improve the flow of the manuscript. With this in mind, we choose to leave these figures in the Supplementary.

Line 421. While this statement may hold true, comparing the tail of the size distributions is not the same as calculating the value of CN3-10. While you do not have CN10, you could still calculate CN3-8, which is a similar comparison. Good suggestion, thank you. Further analysis has been completed at this size fraction. As might be expected, the CN3-8 fraction is most highly concentrated in the 'Polar' measurements. In terms of magnitude the results are similar, however they are fractionally quite different.

| Median concentrations | Polar | Ferrel |
|-----------------------|-------|--------|
| $CN_3 (cm^{-3})$      | 594   | 265    |
| $CN_{3-8} (cm^{-3})$  | 148   | 118    |
| CN3-8:CN3             | 0.25  | 0.45   |

This suggests a greater proportion of fine aerosol in the total population in the Ferrel cell measurements, despite these measurements being significantly less concentrated. I have added this information to the text.

**Minor comments/suggestions:**

*The manuscript formatting does not follow the ACP requirements.* – I have done my best here and used the Copernicus template, though recognise the references font is too small. Are there other areas I need to change?

Line 56. Change 'far' to 'remote' – Thank you, this has been changed.

Line 121. add space between 'L min-1' - Thank you, this has been changed.

*Figure 2. Maybe define the Ferrel cell wind direction as WNW to be consistent with the text. The width of Ferrel cell sector is 73 deg, not 68 deg.* – Thank you, this has been changed.

Line 202. What are 'these' measurements - Thank you, this has been clarified.

Line 207 15% of figure 2a or 2b? - Thank you, this has been clarified.

*Line 211. I know you defined the bin width elsewhere, but it may be worth restating here.* – Thank you, this has been clarified.

Line 212. Stick with 'sectors' and remove 'windows' - Thank you, this has been changed.

Line 223. The measurements are labelled "Southern" in Figure S1 (not "No Category")- Thank you, this has been changed.

Line 282 Reference the relevant figure – Thank you, this has been changed.

Line 292 What finding? - Thank you, this has been clarified.

Line 299 Reported where?- Thank you, this has been clarified.

*Line 334 It would be better to explain what a 'Type 3 and Type 4' day is in Chambers et al. (2018)* – As mentioned earlier, this section has been edited for improved clarity.

*Figure 6. change atmospheric pressure to atmospheric sea level pressure* – Thank you, this has been changed.

Section 5 – Please provide a direct link to the data. – The DOI for these measurements is currently being processed. This link will be provided when the archiving is complete.

---

## Author Response (AR1)

Anonymous Referee #1 Received and published: 21 December 2020

The authors thank referee 1 for their time spent reviewing this manuscript and their valuable advice in suggested improvements. We have edited the manuscript in line with the suggestions from referee 1 as detailed below.

Manuscript updates are visible in the uploaded document containing tracked changes, and noted here. Referee comments are included in this document in italics, and author responses and manuscript changes formatted normally.

*Figure 1: Why are there no observations in one of the north/south transects of the SO? Ship exhaust? Or is it the software malfunction?* Yes, referee 1 is correct: measurements were not taken on one transect due to software malfunction. This has been clarified in the figure caption.

Manuscript changes: Update of Figure 1 caption (lines 208-214).

**... results on the other half of the measurements would be nice to provide a more complete picture, even if the air mass origin is not as precisely known...**

Thank you for this suggestion. Total-voyage plots for  $CN_3$ ,  $CCN_{0.55}$  and 222Rn have been included in Figure 4, along with the 'Southern' and 'No category' wind classifications. A total voyage median size distribution can now be found in the supplementary. I have also added a paragraph describing the overall voyage observations at the start of the results section. We agree that this adds useful depth to the manuscript.

Manuscript changes: Inclusion of total-measurement statistics in Figure 2 (previously Fig. 4), Figure S1, supplementary tables and in text (lines 339-364).

**Why might transit aerosol (CN3, CCN0.55, distribution modes) be higher than the Ferrel cell category if the transit is largely in the Ferrel Cell?**

Thank you for this comment. The transit category of measurements represents a 'classic' remote marine bimodal aerosol distribution and encompasses the transit of the Investigator back to Hobart, whereas the 'Ferrel' category is representative of the southern extent of this regime where greater influences from, for example, biological secondary aerosols is expected. It is also likely that measurements from the Transit period of the voyage are influenced by continental and anthropogenic aerosol sources at their northern extent. In the 'Ferrel' category, these sources have little to no influence on local aerosol population as measurements are taken below 56°S. It must also be mentioned that homogenous aerosol concentrations in such large-scale atmospheric circulations are not expected- rather local sources and sinks will have a greater impact. I have clarified this and added an additional reference.

Manuscript changes: Additional detail with respect to Ferrel/Transit populations (lines 546-550).

Line 298-309 It is a little hard to follow this paragraph, but more importantly, it's unclear what the point of this paragraph is. If you are going to point out differences in mode diameters, some reasoning as to why they are different is necessary to justify the importance

of identifying them. I am not sure you can provide some context for some of these small differences between mode diameters, but without it, the figure alone is enough to show the differences.

Line 343-352 Similar to the previous comment, you are essentially describing what is seen in the figures, but not providing context on why these differences are important or what they mean.

Upon reflection, we agree this paragraph is difficult to understand and adds little useful detail to the manuscript. There it has been removed, (along with the similar paragraph at lines 343-352) and replaced with the following text.

"Significant differences between the modal size bins do exist between categories when calculating the uncertainty using this method. However, these differences are small (less than 12 nm in each case) for both wind and AH categorisations. It is therefore difficult to draw any conclusions from this result."

Manuscript changes: Above text added (lines 605-609).

Line 318 – 339 Much of this comparison to Chambers et al. 2018 requires the reader to read Chambers et al. 2018 to understand it. It would be helpful to briefly include the necessary details from Chambers et al. 2018 to understand the comparisons you have made so that the reader does not have to read another manuscript to understand yours. Thank you for making this section more comprehensible. More details have been added to this section in an attempt to clarify how the present work compares to Chambers et al. (2018) in order to improve the clarity of the manuscript.

Manuscript changes: Explanation added at line 649.

Section 3 is organized as if sections 3.1-3.4 are results and section 3.5-3.7 is the discussion. If you intend to keep the results and discussion separate, I suggest labelling the sections as such (I.e. change section 3 to results and add a section 4 titled Discussion which would included section 3.5-3.7). Otherwise you leave the reader wondering where the discussion is (as I have shown in my previous comments).- Thank you for this suggestion. The authors believe sections 3.5-3.7 contain most of the discussion, though some results are still being presented in these sections. Therefore, I have decided to leave the structure organised as it is, however, have included a clarifying sentence at the end of Section 3.4 ("The significance of the results presented above will be discussed in the following paragraphs along with other supporting evidence.")

Manuscript changes: Above text added to line 666.

*I suggest moving Figures S2 and S3 to the main text as they provide important context.* – Thank you for this suggestion. The authors are not sure this is the best approach- these figures have been included in the main text in previous versions, but on the advice on multiple co-authors have moved them to the supplementary to improve the flow of the manuscript. With this in mind, we choose to leave these figures in the Supplementary.

Manuscript changes: None.

Line 421. While this statement may hold true, comparing the tail of the size distributions is not the same as calculating the value of CN3-10. While you do not have CN10, you could still calculate CN3-8, which is a similar comparison. Good suggestion, thank you. Further analysis has been completed at this size fraction. As might be expected, the CN3-8 fraction is most highly concentrated in the 'Polar' measurements. In terms of magnitude the results are similar, however they are fractionally quite different.

This suggests a greater proportion of fine aerosol in the total population in the Ferrel cell measurements, despite these measurements being significantly less concentrated. I have added this information to the text.

Manuscript changes: CN3-8 statistics presented (line 756).

**Minor comments/suggestions:**

*The manuscript formatting does not follow the ACP requirements.* – I have done my best here and used the Copernicus template, though recognise the references font is too small. Are there other areas I need to change?

*Line 56. Change 'far' to 'remote'* – Thank you, this has been changed. Manuscript changes: "far" changed to remote (line 80).

*Line 121. add space between 'L min-1'* - Thank you, this has been changed. Manuscript changes: as suggested. Line 121

*Figure 2. Maybe define the Ferrel cell wind direction as WNW to be consistent with the text. The width of Ferrel cell sector is 73 deg, not 68 deg.* – Thank you, this has been changed. Manuscript changes: Figure 2 caption updated.

*Line 202. What are 'these' measurements* – Thank you, this has been clarified. Manuscript changes: clarified.

*Line 207 15% of figure 2a or 2b?* – Thank you, this has been clarified. Manuscript changes: "of figure 3a" added (line 421).

*Line 211. I know you defined the bin width elsewhere, but it may be worth restating here.* – Thank you, this has been clarified. Manuscript changes: Widths restated, lines 449-451.

*Line 212. Stick with 'sectors' and remove 'windows'* – Thank you, this has been changed. Manuscript changes: Sectors in place of 'windows'(line 449).

Line 223. The measurements are labelled "Southern" in Figure S1 (not "No Category")- Thank you, this has been changed. Manuscript changes: Figure S1 labels updated.

*Line 282 Reference the relevant figure* – Thank you, this has been changed. Manuscript edits: reference to relevant Figure (line 460).

*Line 292 What finding?* - Thank you, this has been clarified. Manuscript changes: language clarified, line 477

*Line 299 Reported where?*- Thank you, this has been clarified. Manuscript changes: clarification (line 548).

*Line 334 It would be better to explain what a 'Type 3 and Type 4' day is in Chambers et al. (2018)* – As mentioned earlier, this section has been edited for improved clarity. Manuscript changes: Explanation added at line 649.

*Figure 6. change atmospheric pressure to atmospheric sea level pressure* – Thank you, this has been changed.

Manuscript changes: Figure 6 labels edited.

Section 5 – Please provide a direct link to the data. – The DOI for these measurements has been added.

Manuscript changes: Data availability statement updated.

Anonymous Referee #2 Received and published: 28 December 2020

The authors thank referee 2 for their time and attention in reading this manuscript. The feedback provided by referee 2 has resulted in significant changes to parts of the manuscript, changes which the author believe make this a stronger and more balanced piece of work. Specifically, the significance of the work has been made clearer and the relevance widened. Responses to specific comments are provided below.

Manuscript updates are visible in the uploaded document containing tracked changes, and noted here. Referee comments are included in this document in italics, and author responses and manuscript changes formatted normally.

Overall, the measurements are well described and the analysis is straight forward. However, there are a number of points to be improved: The particle losses in the inlet are not described. It is also not clear whether the same inlet system was used in Humphries et al., which is important to know because this work is based on an intercomparison. In addition, the scientific motivation for this investigation is not well presented. It seems that the mere purpose of this study is to compare against Humphries et al. (2016), which is not much scientific merit per se. Better arguments for the relevance of this study need to be provided to make it publishable. Much more in depth discussion on the significance of the findings is needed. After reading the manuscript the question "so what?" is not answered. In addition, the data were analyzed selectively following a too narrow scientific scope, namely those data that fit the scheme of Ferrel and Polar cell. All data should be presented and discussed in the main text to provide a more comprehensive idea of aerosol in the so scarcely measured Southern Ocean.

Thank you for this feedback. The authors have implemented a number of changes to manuscript in response to these comments. These include:

- More clearly stating the aims of research (lines 204-209)
- Improved discussion of the relevance of these type of measurements in the introduction (lines 54-79)
- Better description of inlet and modelled losses (lines 244-252)
- More comprehensive description of significance of findings (lines 836-852)
- Broadening of scope of analysis (lines 339-383)

**Specific comments:**

Abstract: I recommend rewriting the abstract. It should contain information on what was measured and when, on which platform etc. In the current form, it builds too much on the previous study by Humphries et al. and readers who are not familiar with that study will not get a good insight into what was done in this study. The abstract also heavily relies on knowledge of the Polar and Ferrel cells are, but those are not introduced. I recommend explaining what they are. It is also not clear whether the absolute humidity was measured or obtained from numerical weather prediction models. Furthermore abbreviations are not introduced.

Thank you for this suggestion. The abstract has been re-written as suggested, with a greater focus describing the measurements and main results of the work.

Manuscript changes: re-written abstract (lines 15-53).

Introduction

*l.* 49: It is unclear which model parameters the authors refer to. – Thank you- this sentence has been edited.

Manuscript changes: better explanation added (line 65).

*l.* 55: The argument why understanding the aerosol population over the East Antarctic seasonal ice zone is not very well fleshed out. Only a study of Shindell et al. (2013) is cited which highlights discrepancies between modelling and satellite observations. From this it is however not clear why this should researched further. This argument needs to be elaborated.

Thank you for this suggestion. Detail has been added to this section, and the introduction has been restructured to highlight the motivation more clearly including adding relevant recent references. Thank you specifically for the suggestion of McCoy et al., it is a very relevant publication!

Manuscript changes: addition of detail, references and restructure of first paragraphs of introduction (lines 65-87).

*l.* 60-64: The discussion of results obtained from previous measurements over the Southern Ocean is extremely short and it is unclear what the main point is that the authors would like to make. Only two studies are cited and only one short result of one of the two studies is mentioned. Here, clearly, a more thorough review needs to be done and the main findings of several previous studies relevant to this particular work need to be synthesized.

Thank you for this feedback. We have significantly edited the introduction to include a more thorough review of the relevant literature.

Manuscript changes: addition of results from previous studies (lines 125-200).

*l.* 75 *ff*: Say where the cells roughly meet in terms of latitude. At the end of the introduction, I expect a brief description of the campaign, i.e. when did it happen, what was measured...

Thank you for this suggestion. The authors agree this helps the flow of the manuscript, and as such have added a brief campaign description as suggested.

Manuscript changes: include latitude of Polar-Ferrel cell meeting (line 163). Improved description of voyage in question (lines 201-204)

Methods

*Fig. 1: I recommend plotting the sea ice extent for SIPEX-II an PCAN since this is an important environmental factor for this study.*

Figure 1 has been updated with this change implemented. Thank you (line 208).

L. 110 ff: Meteorological measurements on ships can be biased because of the ship's superstructure (see e.g., Landwehr et al. (2020), 10.5194/amt-13-3487-2020). Did the authors do anything to check for potential biases and were the data corrected? A more detailed description is needed.

No bias correction was applied to the meteorological measurements. However, it is expected periods of meteorological measurements influenced by the ship's superstructure are coincident with periods of exhaust contamination in other parameters, which have been removed. This is due to the relative

position of the anemometer positioned at the bow of the ship, forward of the superstructure and exhaust stack of the *Investigator*. This has been clarified in the text.

Manuscript changes: included detail of meteorological sensor location and filtering (lines 239-243).

*l.* 115: Particles as small as 3 nm can easily get lost through diffusion. There is no reason not to characterize the inlet losses at least theoretically, if experimentally this is not possible. The authors can use for example the particle loss calculator by Von der Weiden et al. (2009), www.atmos-meas-tech.net/2/479/2009/. Also the length of the inlet line is not given and neither are the bends described. This needs to added to the paper. The concentrations reported in this work might be used for satellite and model validation, hence providing uncertainties with the numbers, and those include inlet losses, is essential.

Thank you for bringing this up- this is a very relevant issue. Experimental characterisation of inlet losses on the *Investigator* was planned for 2020 but could not occur. We have added a description of the inlet system along with theoretical inlet transmissions in Section 2.2. SIPEXII was aboard the icebreaker *Aurora Australis*, whereas PCAN occurred on the *RV Investigator*. Therefore, inlet systems used were different and have been characterised separately. This has been clarified in the text.

Manuscript changes: description of *RV Investigator* inlet (lines 244-252). Clarification of SIPEXII voyage platform (line 158).

*l.* 120: How were size distributions measured? One expects this information here, where they are mentioned not further below. – This section has been restructured to improve the clarity. Thank you.

Manuscript changes: Shifted description of SMPS instrument up in the text (lines 276-283).

*l. 140: There is no mention of what Rn measurements were used for and why.-* An explanation of the inclusion of Rn measurements has been added- thank you.

Manuscript changes: justification of RN measurements included (lines 292-294).

*l. 145 ff: How were CO2 and black carbon measured? This information needs to be included in the paper.* – The instrumental techniques have been added to the manuscript, thank you.

Manuscript changes: addition of descriptions of CO2 and BC measurements (lines 294-300).

*l. 160: Say which variables were used along the trajectories and which uncertainties are referred to? Do the authors mean the 3 D coordinates or the uncertainties in meteorological parameters?* More detail and a reference has been added describing trajectory uncertainty.

Manuscript changes: reference to literature regarding uncertainty of HYSPLIT over Antarctica and detail of uncertainty (lines 323-325).

*l.* 169: Explain what the McGill et al. (1978) style is. Most readers will not be familiar with it and will have to look up the reference. – Thank you. A description has been added.

Manuscript changes: additional explanation of CI calculation (lines 334-335).

**Results**

*l.* 175: The question is why one should expect a step change? Just because this was observed previously, once only, it does not mean that this is the norm. The authors should explain why their implicit assumption is the step change, or they should drop it and describe more neutrally that a step change has been observed previously, but one cannot expect it to be there permanently as it will depend on the meteorological situation. The Polar and Ferrel cells are descriptions of longer-term average circulation features. Those can deviate significantly on the shorter term. It is also unclear why it is so important to determine the transition between the two cells in the first place, and in the second place why is it important to use aerosol parameters? Those two points are not very well motivated. We have attempted to change the tone of writing here, and comment more objectively on the aerosol populations observed during the PCAN voyage. It is noted that the distinction between climatological and meteorological phenomena is important in this case, and as such this in mentioned in text. Thank you for your comments.

Manuscript changes: Clarification of language surrounding aerosol changes (373-379).

*Fig. 2: All fonts are way too small. There are also substantial measurements near 180 in Fig. 2b. It is unclear why those are not considered in the analysis if the argument, as given in the caption, is the frequency of observations. Fig. 2c has no y-axis scaling.* Thank you for this comment. I have adjusted this and made the 'Southerly' measurements (near 180°) a more prominent part of the manuscript. I have also included the y-axis on 2C.

Manuscript changes: Increased size of Figure 2 (now Figure 3). Addition of 'Southerly' category in Figures 2 and 3. Increased mention of 'Southerly' measurements (lines 449-452).

*l.* 220ff: Why did the authors exclude measurements from the South? Those are also important and interesting. Just because they do not fit the scheme of Polar and Ferrel cell does not provide a good argument. I recommend data are shown and discussed. Also statistics for "no category" should be shown for completeness.

Thank you for this suggestion. I have added the 'Southern' and 'No category' classifications to Figure 4 and include median size distributions for these categories in the supplementary. The results are also discussed in text at the beginning of the 'Results' section.

Manuscript changes: Holistic discussion of all measurements (lines 339-370). Addition of "Southerly" and "No\_cat" statistics to supplementary tables, and Figure 2.

*l. 241: Please provide the formula in the manuscript.* Thank you for this suggestion. The formula has been included.

Manuscript changes: inclusion of AH formula (lines 505-510).

Fig. 3: The writing is too small. The figure font size has been increased. Thank you.

Manuscript change: Figure 3 (now Figure 4) font size increased. *l. 290ff: It is an interesting result that the accumulation mode diameter in the Polar cell is smaller than that in the Ferrel cell. This is opposite of what Schmale et al. (2019) found for the Western site of Antarctica. The authors should discuss their findings in a*  broader context, going beyond what has been found South of Australia. The question of larger particle number concentrations closer to Antarctica is not limited to that sector of the Southern Ocean. The author very briefly hint at potential causes in l. 485, but do not make the connection as suggested here.

Thank you for suggesting the reference. Comparison to these results from west Antarctica has been added. I do not think the difference in median sizes large enough to be commented upon in this case (this is likely only 1-2 bins of the SMPS at this size). This is a difference, however, compared to Schmale et al., who note a larger difference in accumulation modes in their clusters.

Manuscript changes: Increased comparison to study from west Antarctica (lines 586-598).

*Fig. 5: Diameter axes are typically logarithmic.* The x axis in this case is logarithmic. The caption has been edited to state this.

Manuscript changes: Figure 5 caption edited.

*l.* 334: the Types in Chambers et al. (2018) need to be explained here, otherwise it is not clear what the authors try to convey. – Thank you, this explanation has been added. Manuscript changes: Chambers' 'types' explained (lines 649-652).

*l. 333: Was "SO" introduced?-* No, the acronym has been removed. Thank you for the pickup.

Manuscript changes: SO changed to Southern Ocean (line 670).

*l.370ff:* This is highly speculative. What are the observations that allow the authors to draw the conclusion that the lifetime of aerosols is longer in the Ferrel than the Polar cell? Particle lifetime is strongly determined by precipitation for the considered size range. What are the differences between the cells? How long would it take to increase the size of an accumulation mode particle if the authors assumed it is sulfuric acid with a given accommodation coefficient? How long does coagulation take? Is this in line with the expected longer lifetime? Also, if the source is similar in both cells, why are there less particles in the Ferrel cell? The most likely explanation is precipitation scavenging, this would however counteract the lifetime argument. This statement clearly needs more elaborate discussion to back up the assumption. Thank you for this suggestion- in fact this thought was rather frivolous, without the required analysis. It has been removed.

Manuscript changes: Speculative suggestion of aging distributions removed (line 700).

*l. 390: What are the vertical lines in figures S2b and S3b?* – These represent the interquartile range of the trajectories. I have added this to the caption.

Manuscript changes: Supplementary Figures S2, S3 captions edited.

Conclusion:

*l.* 515: Why does it need strong low pressure systems to exchange air masses between the free troposphere and the marine boundary layer? Smaller convective processes during cloud formation and dissipation will do the same. Here more discussion is needed on how air masses might be exchanged. Thank you for reminding us of this fact. Low pressure systems are mentioned specifically as there is anti-correlation between particle concentration and surface pressure observed in the measurements. I have added a sentence immediately below regarding smaller scale exchanges.

Manuscript changes: detail about air exchange on less than synoptic scales added (lines 816-821).

*l.* 520: This work does not provide evidence for a seasonal cycle. It merely compares summer to spring measurements. There is no information on the fall and winter yet.- Agreed- this sentence was poorly phrased. It has been edited.

Manuscript changes: 'seasonal cycle' changed to 'seasonal differences' (line 870).

*Technical remarks: l. 128: Cloud condensation nuclei number...* – This has been edited. Thank you.

Manuscript changes: 'CN' changed to 'CCN'.

---

## Author Response (AR2)

**Author Response to Referees: Revised Submission**

acp-2020-1213

Summer aerosol measurements over the East Antarctic seasonal ice zone

Jack Simmons et al.

**a) Anonymous Referee #1**

The authors wish to thank referee #1 for their time in reviewing the revised manuscript. The authors are grateful that referee #1 finds our changes to the initial manuscript satisfactory. Comments on the revised submission are addressed below. The referee is quoted in italics, and author responses in standard font. Edited text from the manuscript is included in quotations. A document showing tracked changes since review is also included in the revised resubmission.

*I believe the authors have made satisfactory corrections to the manuscript and responses to the reviewers. The only minor issue I found was an incomplete sentence and repeated sentences in the Figure 1 caption.*

Thank you for this feedback. The caption for Figure 1 now reads:

"Figure 1: Voyage tracks of the SIPEX-II and PCAN voyages. Voyage tracks of the SIPEX-II (spring 2012, grey) and periods of the PCAN voyage (summer 2017, orange) for which measurements were analysed. Measurements from the southward PCAN transit are excluded due to instrument malfunction. Grey shading represents the sea ice area observed at the mid-point of the SIPEXII voyage as detected by the Nimbus-7 satellite (National Snow and Ice Data Centre, 2021). No sea ice was present in the plotted region at the midpoint of the PCAN voyage. An interactive map of RV Investigator voyage tracks can be found here: https://www.cmar.csiro.au/data/underway/?survey=in2017_v01."

*In the future, it would help the reviewer if the authors included the changes made to the text and figures in the reviewer responses when practical.*

Thank you for this feedback- this is understood and this information will be included from now on.

a) Anonymous Referee #3

The authors wish to especially thank referee #3 for reviewing this manuscript as this stage of proceedings. Their comments have proved useful and the authors believe these points serve to clarify the message of the manuscript. As for referee #1's comments, responses to comments on the revised submission are addressed below. The referee is quoted in italics, and author responses in standard font. Edited text from the manuscript is included in quotations. A document showing tracked changes since review is also included in the revised resubmission.

**Main comments:**
*The structure of the discussion must be revised. The description of categorization by wind direction and absolute humidity should be part of the Methods section. Another option is to start the Results and Discussion section with this description. Lines 223-292 should be moved further down, after the description of the categorization method. Figure 2 does not make any sense when referred to and discussed before this description. My recommendation is to start with the description of this method, and then, to present and discuss aerosol results.*

Thank you for this feedback. As suggested, the text in this section has been re-arranged. The following text has been inserted to the start of the results and discussion:

"In the following sections, categorisation of air masses has been performed using both wind direction and absolute humidity. Following a definition of the borders of the categories, the overall aerosol properties observed during the voyage are presented. Differences in aerosol properties within categories defined by wind direction and absolute humidity are then considered and the implications of the findings discussed."

Consideration of the overall aerosol properties observed during the campaign (previously lines 223-292) has been shifted after the description of the categorisations. This now exists in the manuscript as:

"**3.3 General properties of observed aerosol populations**"

All sections in the Results and Discussion have been re-numbered to account for the addition of this new section (i.e. what was previously Section 3.3 is now 3.4 etc.). Also, Figure 2 has been shifted below Figure 4, and figure numbers (2-4) adjusted accordingly. In-text figure references have also been edited.

*SIPEX-II / PCAN comparison:*
*1. while the authors highlight the difference in sea-ice extent in Figure 1, this is not mentioned later in the discussion. Could the different environmental conditions explain part of the differences during the two campaigns?*

This is a very useful revision- thank you for the suggestion. The following text has been added in Section 3.8 (Explaining observed seasonality).

"Spring measurements taken during SIPEXII voyage were taken while significant concentrations of sea ice surrounded the measurement platform, as indicated in the shading in Fig. 1. Very limited sea ice was present in these latitudes during the PCAN voyage. It is important therefore to consider the implications of ice-melt related algal bloom when comparing spring and summer measurements in the seasonal ice zone."

*2. In addition, the authors do not fully address the point made by reviewer #2 about the description of the inlet system. The description of the sampling inlet and associated inlet losses on the Investigator are a welcome addition to the manuscript. However, it does not explicitly tell how the two systems compared. The authors mentioned in their response that the inlet system used on Aurora Australis was characterized separately. A few lines about that in the Methods section and later in the discussion would be appreciated.*

Thank you for this feedback, we have added some detail in comparing the inlet systems. The following text has been added to the methods section, Section 2.2.:

" The present work includes comparison to aerosol concentrations reported by Humphries et al. (2016) taken on board a different research vessel, the Aurora Australis. It is prudent at this stage to note the differing inlet efficiencies for this separate inlet system. An overall inlet transmission efficiency of 0.89 has been reported, generated from a combination of theoretical and experimental characterisation (Humphries et al., 2015)."

The following has also been added to the Results and discussion, Section 3.7.

"It is important to note that these measurements were taken from different platforms, and therefore different particle inlet transmission efficiencies must be considered. However, the difference in the magnitude of enhancement observed in the Polar cell, which does not depend on inlet transmission efficiency in both voyages demonstrates that there is a robust difference in aerosol populations observed between voyages."

*3. Lines 549-551: Did Humphries et al. 2016 use the same method to categorize measurements? It is unclear to me if that's the case and thus how concentrations in Polar cells during the two campaigns can be directly compared.*

It is noted that different techniques were used to classify Polar cell measurements between voyages – SIPEXII used changes in aerosol populations backed up by trajectory analysis. Insufficient immediate change in aerosol population was observed in the PCAN measurements for the same technique to be used. Therefore meteorological variables were used in this study to define the cells. It is recognised that these techniques differ. However, the Polar cell/Ferrel cell division is climatological and therefore it can be expected that dominant wind directions associated with measurements in each cell are persistent across seasons. Also, back trajectory analysis for the PCAN measurements divided by wind direction shows a favourable comparison with those from SIPEXII (free tropospheric influence, air masses travelling from above the Antarctic continental interior before descending near the Polar front for Polar cell measurements compared to transport in the marine boundary layer for Ferrel cell air masses.). Text has been added at this point (Section 3.7) to clarify this:

"Humphries et al. (2016) defined Polar cell measurements using a clear change in observed aerosol population along with detailed back trajectory analysis. This method is different to that used here. However the persistent nature of dominant surface winds associated with Polar and Ferrel cells, coupled with back trajectories (plotted in supplementary Fig.S2 and Fig S3) aligning with those modelled for SIPEXII measurements allows comparison between Polar cell measurements to be made between the voyages."

*In the abstract, and later in the discussion, the authors mention "seasonal differences in atmospheric circulation including potentially more significant katabatic outflow in summer". How robust is this assumption? According to Parish and Cassano (2003), "summertime winds cannot be expected to contain a significant katabatic component, owing to enhanced solar heating of the ice slopes".*

Thank you for this comment- you highlight a flaw in our reporting. This should be explained better: we propose increased influence of katabatic flow on observed aerosol population, driven by weaker and less frequent synoptic low-pressure systems (compared to spring). This has been clarified in the abstract:

"However, the smaller changes observed in the present analyses suggest seasonal differences in atmospheric circulation including lesser impact of synoptic low-pressure systems in summer."

The relevant text in the discussion has also been changed (Section 3.8), and now reads:

"Instead, **increased influence of the katabatic outflow** on measurements in the Polar cell documented by Chambers et al. (2018) is observed and strengthens the relationship between CN3 concentration and AH. It is recognised that katabatic outflow has an annual minimum in summer (Parish and Cassano, 2003). However, during periods of limited insolation, downslope drainage driven by latent cooling still occurs during the summer months, especially in the high-wind East Antarctic (e.g. Yu et al., 2020, Chambers et al., 2018)."

Finally, the wording of the conclusion has also been edited, and now reads:

"Instead, katabatic outflow from the Antarctic continent more strongly influences particle number concentration enhancement."

*The authors mention that "southern measurements are separated from the Polar/Ferrel classification in an attempt to isolate the known influence of katabatic outflow on near-coast measurements". While I fully support this initiative, Table S2 shows that there's more measurements with low AH in the polar vs. southerly category. I'm concerned this method is not entirely successful here.*

It is recognised that using wind direction is not a perfect way to identify katabatic outflow. Using, for example, diurnal cycles in $Rn^{222}$ as done by Chambers et al. (2018) is a more robust method.

However given the context of classifying measurements by wind direction, this method was selected for consistency in this manuscript. Humphries et al. (2016) proposed an atmospheric transport pathway of air mass descent from the free troposphere transport pathway in the polar cell in this region. This transport pathway was also observed in 'continental Antarctic airmasses' by Alroe et al. (2019). It would be expected that this air, having spent significant time in the free troposphere, would be associated with low levels of absolute humidity. Figure S2 provides evidence for this, with many of the airmasses classed as 'Polar' showing recent evidence of free tropospheric residence.

Also, given that katabatic flow is not dominant all of the time (as suggested earlier) during the Antarctic summer, the frequency of katabatic flow is expected to be relatively small. From section 3.6:
"It should be noted that the Low AH category includes a greater percentage of the measurements (45%) compared to the Type 3 and 4 days reported by Chambers et al. (2018) (~27%) so likely includes measurements not influenced by katabatic drainage."

Therefore it is expected that significant overlap exists between the Polar and Southerly categories. The follow sentence has been added:
"… southern measurements are separated from the Polar/Ferrel classification in an attempt to isolate the known influence of katabatic outflow on near-coast measurements. It is recognised that identifying katabatic flow by wind direction is not ideal. However, given the context of classifying measurements by wind direction, this method has been used in this study for consistency."

*I found a number of typos (see line-by-line comments), a thorough proofreading step would have been appreciated.*

The authors appreciate this feedback and will take this into account in future work.

**Line-by-line comments:**
*Line 21: Please indicate where the cruise started, something like: "(…) during a summer 2017 voyage from Hobart, Tasmania, to the East Antarctic seasonal sea ice zone".*

Thank you- this edit has been made as suggested. The text now reads: "…during a summer (January-March) 2017 voyage from Hobart, Australia to the East Antarctic seasonal sea ice zone."

*Figure 1, caption: the sentence "Measurements from the southward transit are excluded due to instrument malfunction" is mentioned twice (line 139-140 and line 140-141). In addition, a portion of the sentence is missing and we don't know what is supposed to indicate the*

*voyage period for which measurements were analyzed. In addition, could you add ground-based stations on the map, e.g., Casey and Dumont d'Urville, to help the readers orientate themselves?*

Thank you- the caption has been adjusted and duplication removed. The caption now reads:

"Figure 2: Voyage tracks of the SIPEX-II and PCAN voyages. Voyage tracks of the SIPEX-II (spring 2012, grey) and periods of the PCAN voyage (summer 2017, orange) for which measurements were analysed. Measurements from the southward PCAN transit are excluded due to instrument malfunction. Grey shading represents the sea ice area observed at the mid-point of the SIPEXII voyage as detected by the Nimbus-7 satellite (National Snow and Ice Data Centre, 2021). No sea ice was present in the plotted region at the midpoint of the PCAN voyage. Nearby Antarctic stations are plotted to assist interpretation. An interactive map of RV Investigator voyage tracks can be found here: https://www.cmar.csiro.au/data/underway/?survey=in2017_v01."

Figure 1 has also been updated, including the location of the stations mentioned.

*Line 228-229: "it has been necessary to limit this analysis has been limited to the time period of the shortest dataset". Please rephrase.*

Edited as suggested, sentence now reads: "…it was necessary to limit this analysis to the time period…"

*Line 274-275: "A median size distribution plotted (….) has a bimodal shape". Are the authors referring to a specific figure here? See main comment above on the structure of the discussion, it is currently quite confusing.*

Thank you, understood. The sentence now reads: "A median size distribution plotted from all valid SMPS measurements for the analysis period during PCAN (supplementary Figure S1,black trace) has a bimodal shape…"

*Line 276: typo "close to 70 nm.."* Thank you- the extra full stop has been removed.

*Figure 2: I would replace "No_cat" by "Other". "No_cat" seems to suggest missing wind data, which is not the case. Also please give more detail in the caption, e.g., explain what the low, mid, and high AH categories refer to (give the thresholds in gH2O/m3). The caption should be self-explanatory. Finally, can you use different colors for "No_cat" and "southerly"? (I know I'm being picky here).*

Thank you for this comment. The name of the 'No_cat' classification has been altered to "Other" as suggested in figures and in text. The colour palette has also been slightly adjusted to ensure two distinct categories are not plotted using the same colour. This change in colour palette and category names is also reflected in Fig. 2 and Fig. S1.

The order of figures has now been changed, so the box plots (previously Figure 2) are now included further down the manuscript than the frequency histogram of AH measurements (previously Figure 4). The caption of the frequency histogram of AH measurements details the boundaries, as does the caption of the wind rose figure (previously Figure 3, now Figure 2) for the wind measurements. Hopefully this restructure means readers are able to interpret the boxplots correctly without restating category boundaries in the caption.

*Figure 3: same as above, the caption should be self-explanatory. Explain what the 56.6 S latitude threshold is, something like: "Wind rose for wind measurements south of 56.6 S, i.e., excluding homebound transit to Hobart, Tasmania". In addition, "Frequency histogram of wind measurements plotted in 2b", should be 3b.*

Thank you for this comment. The caption has been edited and now reads:

"Figure 2 (a): Wind rose for the entire analysed PCAN voyage (February 6-March 4, 2017). (b): Wind rose for wind measurements south of 56.5° S (excluding measurements from the homeward transit to Hobart Tasmania, the Transit category), showing selected sectors. There are three areas demonstrating a higher density of wind measurements, from the WNW (Ferrel, sector: 225-298°, width 73°), from the SE (Polar, sector 99-148°, width 49°) and from the S (Southerly, 148-217°, width 69°) . These sectors, capturing 76% of available wind measurements, are shaded in blue (Ferrel, 22%), grey (Southerly, 20%) and orange (Polar, 34%) respectively. (c): Frequency histogram of wind measurements plotted in 2b with 1° bin size. Ferrel (blue), Southerly (grey) and Polar (orange) sectors are highlighted."

*Lines 321-323: "The southern edge of the Ferrel cell is expected to display a dominant NW wind, and the northern edge of the Polar cell is expected to be associated with a dominant SE wind". Can you cite a few papers here to back this up?*

I have added some citations- a mix of articles and reference texts.

*Line 447: typo. "in the differences in difference".*

Thank you, this sentence now reads "… are evident in the differences in observed particle concentrations…"

*Figure 5, Fig S2 and S3: I'd appreciate having the southerly category plotted here as well.*

Thank you- we appreciate the usefulness of this suggestion. However, the authors experimented and found that plotting > 3 distributions on a single panel makes it very difficult to interpret the plot. The median size distribution for the southerly category is therefore plotted in Figure S1 instead of in Figure 5 as requested.
The Southerly and 'Other' categories have been included in Figure S2 as requested. Text noting this has been added to Section 3.6:

"Interestingly, the trajectories for the Southerly category also show history of descending airmasses."